# An Evidence-Based Post-Hoc Adjustment Framework for Anomaly Detection Under Data Contamination

**Sukanya Patra**
Department of Computer Science
University of Mons
Mons, Belgium
sukanya.patra@umons.ac.be

**Souhaib Ben Taieb**[*]
Department of Statistics and Data Science
Mohamed bin Zayed University of Artificial Intelligence
Abu Dhabi, United Arab Emirates
souhaib.bentaieb@mbzuai.ac.ae

## Abstract

Unsupervised anomaly detection (AD) methods typically assume clean training data, yet real-world datasets often contain undetected or mislabeled anomalies, leading to significant performance degradation. Existing solutions require access to the training pipelines, data or prior knowledge of the proportions of anomalies in the data, limiting their real-world applicability. To address this challenge, we propose EPHAD, a simple yet effective test-time adaptation framework that updates the outputs of AD models trained on contaminated datasets using evidence gathered at test time. Our approach integrates the prior knowledge captured by the AD model trained on contaminated datasets with evidence derived from multi-modal foundation models like Contrastive Language-Image Pre-training (CLIP), classical AD methods like the Local Outlier Factor or domain-specific knowledge. We illustrate the intuition behind EPHAD using a synthetic toy example and validate its effectiveness through comprehensive experiments across eight visual AD datasets, twenty-six tabular AD datasets, and a real-world industrial AD dataset. Additionally, we conduct an ablation study to analyse hyperparameter influence and robustness to varying contamination levels, demonstrating the versatility and robustness of EPHAD across diverse AD models and evidence pairs. To ensure reproducibility, our code is publicly available[2].

## 1 Introduction

Anomaly detection (AD) is the basis of many critical applications, including cybersecurity (Xiao et al., 2024; Li et al., 2023a), healthcare (Bijlani et al., 2024; Huang et al., 2024), and industrial maintenance (Schwarz et al., 2025; Patra et al., 2024). By enabling the identification of abnormalities, potential threats, or critical system failures, AD contributes to the robustness and safety of real-world systems. Despite its significance, AD remains a challenging task due to the inherent difficulty in characterising anomalous behaviours and the lack of prior knowledge about anomalous samples (Ruff et al., 2021). Consequently, AD is commonly approached as an unsupervised representation learning problem without access to labelled anomalies (Batzner et al., 2024; You et al., 2022).

A standard approach in unsupervised AD involves training a model to learn a "compact" representation of the normal samples from a training dataset under the assumption that the training data is "clean", i.e. contains only normal samples (Ruff et al., 2021). Then, anomalies are identified as deviations from this learned normality. One-class (OC) classification methods (Ruff et al., 2018; Tax and Duin, 2004) learn a decision boundary that encompasses all the normal samples. In contrast, density-based methods (Gudovskiy et al., 2022; Yu et al., 2021) learn the probability distribution of normal samples.

---

[*]Affiliated with the Department of Computer Science, University of Mons.
[2]https://github.com/sukanyapatra1997/EPHAD

39th Conference on Neural Information Processing Systems (NeurIPS 2025).

Furthermore, memory bank-based approaches (Roth et al., 2022) store the features corresponding to normal samples in a memory bank. However, real-world datasets are often contaminated with undetected anomalies (Das et al., 2025; Hien et al., 2024; Qiu et al., 2022). For example, a dataset collected for industrial maintenance may already include unnoticed defects. This leads to biased AD models that struggle to reliably distinguish between normal and anomalous instances.

We consider the more realistic setting where the training data may be contaminated with anomalies. Existing approaches to handle contamination in the *unsupervised* setting primarily follow two strategies. The first employs an auxiliary OC classifier to filter out suspected anomalies (Yoon et al., 2022; Jiang et al., 2022), while the second modifies the training pipeline to enhance robustness against contamination (Qiu et al., 2022; Eduardo et al., 2020). Although effective, these methods rely on prior knowledge of the proportion of anomalies in the training data, i.e. the contamination ratio, which is typically unknown. Also, such methods are often computationally expensive. In the *semi-supervised* setting, methods leverage additional labelled datasets containing normal and anomalous samples (Hien et al., 2024; Ruff et al., 2020). However, their effectiveness diminishes when the anomalous instances encountered during training do not replicate real-world anomalies (Perini et al., 2025).

In this work, we aim to mitigate the possible adverse effects of data contamination on the performance of *unsupervised* AD models (Bouman et al., 2024). Specifically, we address the challenging setting in which training pipelines, data, or prior knowledge of the proportions of anomalies cannot be accessed. This scenario reflects the growing trend of deploying proprietary AD models in real-world applications, where access to internal model components is often restricted. Even when fine-tuning is permitted, it is not only computationally intensive but also unreliable due to the absence of guaranteed clean training data, as anomalies are inherently unknown a priori. This setup aligns with preparation-agnostic *test-time adaptation* (TTA) methods (Karmanov et al., 2024; Zhang et al., 2023; Xiao and Snoek, 2024), which remain largely unexplored in the context of AD. To address this gap, we introduce the **E**vidence-based **P**ost-**H**oc Adjustment Framework for **A**nomaly **D**etection (EPHAD), a simple yet effective method that adjusts the outputs of a pretrained AD model post-hoc, using evidence collected at test time.

Notably, we establish conceptual links between EPHAD and recent advances in test-time alignment for generative models (Mudgal et al., 2024; Li et al., 2024; Korbak et al., 2022), underscoring its broader significance. EPHAD is flexible and can incorporate various forms of evidence, including foundation models like Contrastive Language–Image Pre-training (CLIP) (Zhou et al., 2024; Jeong et al., 2023), classical AD methods such as Local Outlier Factor (LOF) (Breunig et al., 2000), and domain-specific knowledge. Our core contributions are summarised below:

- We introduce EPHAD, a *simple yet effective* TTA framework for unsupervised AD models trained on contaminated datasets. Unlike existing approaches, it requires no access to training pipelines, data or prior knowledge of the proportions of anomalies in the data, making it highly practical for real-world deployments.
- EPHAD performs TTA by combining the prior knowledge captured by the AD model trained on the contaminated dataset and an evidence gathered at test-time. This principled formulation allows for conceptual connections to recent test-time alignment techniques in generative modelling.
- We illustrate the intuition behind EPHAD using a carefully designed toy example. Furthermore, extensive experiments across eight visual AD, twenty-six tabular AD datasets, and a real-world industrial AD dataset demonstrate the effectiveness of EPHAD across diverse unsupervised AD models, evidence pairs.

## 2 Related work

**Unsupervised AD**. Over the years, numerous approaches have been developed for unsupervised AD, which can be broadly categorised into four main families: *one-class classifiers* (OCCs), *feature embedding-based*, *density-based*, and *reconstruction-based* methods. OCCs aim to learn a decision boundary that encapsulates all normal samples. Classical OCC approaches employ shallow models such as support vector-based methods that learn a maximum-margin hyperplane (Schölkopf et al., 2001) or a hypersphere (Tax and Duin, 1999). To mitigate the limitations of manual feature engineering and extend to high-dimensional data, deep learning-based variants like DeepSVDD (Ruff et al., 2018) have been introduced. *Feature embedding-based* methods, on the other hand, leverage

pre-trained deep models to extract representations of input data. These representations are then either stored in a memory bank (Roth et al., 2022; Lee et al., 2022) or used to train a student-teacher network (Zhang et al., 2024; Batzner et al., 2024; Patra and Ben Taieb, 2024). *Density-based* methods detect anomalies by estimating the probability distribution of normal samples, assuming that anomalies reside in low-density regions. While early methods include KDE (Kim and Scott, 2012), more recent deep-learning-based variants include DAGMM (Zong et al., 2018), CFLOW (Gudovskiy et al., 2022), and FastFlow (Yu et al., 2021). Lastly, *reconstruction-based* approaches learn to map normal samples into a lower-dimensional bottleneck and reconstruct them. The inability to accurately reconstruct samples during inference serves as a detection criterion. For a more comprehensive survey, we refer readers to Liu et al. (2024) and Ruff et al. (2021).

**Data contamination**. Handling dataset contamination in AD typically assumes a low proportion of anomalies, allowing methods to prioritise normal instances (inlier priority) (Wang et al., 2019). However, in practice, this assumption is difficult to ensure since anomalies are often unknown. To mitigate contamination, Yoon et al. (2022) proposed a data refinement approach using an ensemble of one-class classifiers (OCCs) to filter suspected anomalies and create a cleaner dataset. While effective, this method incurs high computational costs and discards anomalies rather than leveraging them for improved generalisation via Outlier Exposure (Hendrycks et al., 2019). To address this, Qiu et al. (2022) introduced Latent Outlier Exposure (LOE), which iteratively assigns anomaly scores and infers labels using block coordinate descent while incorporating the contamination ratio to prevent degenerate solutions. However, estimating the contamination ratio remains a challenge. Perini et al. (2022) tackled this by leveraging an auxiliary dataset with a known contamination ratio, assuming domain similarity. Alternatively, Perini et al. (2023) fits a Dirichlet Process Gaussian Mixture Model to anomaly scores, though this approach lacks a closed-form solution. Despite these advancements, existing methods introduce computational overhead and are often impractical for modern pre-trained proprietary models, limiting their real-world applicability.

## 3   Background

Let $X \in \mathcal{X}$ and $Y \in \mathcal{Y}$ denote a pair of random variables following a joint probability distribution $P_{X,Y}$ over the space $\mathcal{X} \times \mathcal{Y}$, where $\mathcal{X} \subseteq \mathbb{R}^d$ and $\mathcal{Y} := \{-1, +1\}$. Here, $Y = +1$ corresponds to the normal class, while $Y = -1$ represents the anomalous class. The conditional distribution of normal samples is $P_{X|Y=+1}$ denoted as $P_X^+$ with PDF $f_X^+$. Likewise, the conditional distribution of anomalous samples is $P_{X|Y=-1}$ denoted as $P_X^-$, with PDF $f_X^-$. The training dataset $\mathcal{D}_{\text{train}}^+ := \{x_i\}_{i=1}^m$ contains only normal samples (uncontaminated) i.e., $x_i \overset{\text{iid}}{\sim} P_X^+$. We denote the test dataset as $\mathcal{D}_{\text{test}} := \{(x_i, y_i)\}_{i=1}^n$ which contains both normal and anomalous samples i.e, $(x_i, y_i) \overset{\text{iid}}{\sim} P_{X,Y}$.

**Density-based anomaly detection**. An anomaly can be defined as "an observation that deviates significantly from some concept of normality" (Ruff et al., 2021). This definition comprises two key aspects: the *concept of normality* and the *significant deviation* from it, which can be formalised using a probabilistic framework. The *concept of normality* is defined as the probability distribution of normal samples $P_X^+$. To formalise this further, we adopt the *concentration assumption* (Steinwart et al., 2005), which posits that although the data space $\mathcal{X}$ is unbounded, the high-density regions of $P_X^+$ are bounded and concentrated. In contrast, $P_X^-$ is assumed to be non-concentrated (Schölkopf and Smola, 2002), and is often approximated by a uniform distribution over $\mathcal{X}$ (Tax, 2001). Given the PDF $f_X^+$ associated with $P_X^+$, which we refer to as *inlier density*, a data point $x \in \mathcal{X}$ is identified as an anomaly if it *deviates substantially* from this concept of normality, i.e., if it resides in a low-probability region under $P_X^+$. However, since $f_X^+$ is typically unknown in practice, density-based methods approximate it using a density estimator.

**Score-based anomaly detection**. Density estimation poses significant challenges, particularly in high-dimensional spaces or when data is sparse, and often incurs substantial computational cost. Fortunately, in the context of anomaly detection, the goal is typically not to recover the exact data likelihood but rather to establish a ranking of data points based on their degree of normality. This motivates an alternative strategy: learning an *anomaly score function* $s_{\text{out}}(x) : \mathcal{X} \to \mathbb{R}$, which directly assigns an anomaly score to a data point $x \in \mathcal{X}$, thereby quantifying its *degree of anomalousness* (Ruff et al., 2021). To complement this, the *inlier score function* is defined as $s_{\text{in}}(x) = -s_{\text{out}}(x)$, capturing the *degree of normality*, where higher values indicate that $x$ is normal. For AD, first, we

train a model to learn the anomaly score function $s_{\text{out}}^+(x)$ using $\mathcal{D}_{\text{train}}^+$. Then, we define the anomaly detector as

$$g_{\lambda_s}(x) = \begin{cases} +1, & \text{if } s_{\text{out}}^+(x) \leq \lambda_s \\ -1, & \text{if } s_{\text{out}}^+(x) > \lambda_s \end{cases} \tag{1}$$

where $\lambda_s \geq 0$ is a pre-determined threshold (Perini et al., 2023, 2022). The density-based AD method can also be interpreted as a specific case of the score-based AD methods where the anomaly score $s_{\text{out}}^+(x) = -\phi(f_X^+(x))$ and $\phi(\cdot)$ is an order-preserving transformation chosen to be the logarithm.

**Data contamination**. For training the AD method, a common assumption is that the training dataset $\mathcal{D}_{\text{train}}^+$ consists solely of i.i.d. samples from the normal data distribution $P_X^+$, without anomalies. However, this assumption is rarely satisfied in practice, since anomalies are typically unknown *a priori*. As a result, the training dataset is often contaminated with undetected anomalies. A more realistic assumption is that the dataset $\mathcal{D}_{\text{train}}^\pm := \{x_i\}_{i=1}^m$ contains both normal and anomalous samples drawn from a mixture distribution $P_X^\pm$ with PDF $f_X^\pm$ (Huber and Ronchetti, 2011; Huber, 1992). Letting $\epsilon = \mathbb{P}(Y = -1)$ denote the *contamination factor*, $P_X^\pm$ can be written as

$$P_X^\pm = \epsilon P_X^- + (1 - \epsilon) P_X^+. \tag{2}$$

As $\epsilon$ increases, the model trained on $\mathcal{D}_{\text{train}}^\pm$ becomes biased towards the anomalous regions, reducing its ability to separate normal from anomalous samples (Qiu et al., 2022; Yoon et al., 2022). The existing literature examining the impact of contamination on unsupervised AD methods (Jiang et al., 2022; Qiu et al., 2022; Hien et al., 2024; Perini et al., 2023, 2022) typically considers contamination levels ranging from 0% to 20%. Additionally, an analysis of 57 datasets spanning Natural Language Processing and Computer Vision in ADBench (Han et al., 2022) [Appendix B.2, Figure B1] revealed that nearly 70% of the datasets exhibit anomaly ratios below 10%, with a median of 5%.

## 4  EPHAD: An evidence-based post-hoc adjustment framework

We consider the realistic scenario in which an AD model has already been trained on a possibly contaminated dataset $\mathcal{D}_{\text{train}}^\pm$. Instead of retraining the model, our goal is to adapt its test-time predictions to mitigate the impact of contamination. To this end, we introduce a novel **E**vidence-based **P**ost-**H**oc **A**djustment Framework for **A**nomaly **D**etection (EPHAD), that corrects model predictions using an evidence function at test-time. The *evidence function* $T(x) : \mathcal{X} \to \mathbb{R}$ assigns higher values to samples deemed more likely to be normal and can incorporate domain-specific knowledge. Thus, EPHAD aligns with *preparation-agnostic* TTA methods (Xiao and Snoek, 2024).

For density-based AD (refer to Section 3), anomalies are identified as samples lying in the low-density regions under the distribution of normal samples $P_X^+$. However, due to data contamination, the trained model estimates the PDF $f_X^\pm$ of the contaminated distribution $P_X^\pm$, as defined in (2), rather than the inlier PDF $f_X^+$. Given an evidence function $T(x)$, EPHAD computes a revised PDF $\check{f}_X^\pm$ using *exponential tilting* as:

$$\check{f}_X^\pm(x) = \frac{f_X^\pm(x) \exp(T(x)/\beta)}{Z_X^\beta}, \tag{3}$$

where $\exp(T(x)/\beta)$ is the evidence scaled by a temperature parameter $\beta \in \mathbb{R}$ and $Z_X^\beta = \int_{\mathcal{X}} f_X^\pm(x) \exp(T(x)/\beta)\, dx$ is the normalising constant. This formulation upweights normal samples according to the evidence while maintaining consistency with the model's original density.

Proposition 4.1 provides a condition under which the revised PDF $\check{f}_X^\pm$ is closer to the inlinear PDF of normal samples $f_X^+$ than the contaminated PDF $f_X^\pm$, in terms of Kullback–Leibler (KL) divergence.

**Proposition 4.1.** *Let $f_X^+$, $f_X^\pm$, and $\check{f}_X^\pm$ be PDFs over the same domain $\mathcal{X}$. Then the KL divergence between $f_X^+$ and $\check{f}_X^\pm$ is strictly less than the divergence between $f_X^+$ and $f_X^\pm$ iff*

$$\mathbb{E}_{x \sim P_X^+}\left[\log \frac{\exp(T(x)/\beta)}{Z_X^\beta}\right] > 0. \tag{4}$$

The proof is provided in Appendix A.1. Consequently, when condition (4) holds, the revised density $\check{f}_X^\pm(x)$ assigns higher relative likelihoods to true inliers, leading to improved separation

between normal and anomalous samples. Hence, we expect EPHAD to yield better anomaly detection performance than the unadjusted model $f_X^\pm(x)$, provided that a suitable detection threshold is used.

Moreover, it can be shown that (3) is the optimal solution to the KL-regularised objective

$$J_{\text{KL}}(\check{f}_X^\pm) := \mathbb{E}_{x \sim \check{f}_X^\pm}[T(x)] - \beta \, \text{KL}(\check{f}_X^\pm \| f_X^\pm). \tag{5}$$

This objective balances two competing goals: aligning the adjusted PDF with the evidence (first term) and maintaining fidelity to the original PDF (second term). The temperature parameter $\beta$ controls this trade-off, recovering the evidence-driven solution as $\beta \to 0$ and reverting to the original model as $\beta \to \infty$. For the proof of (5), see Korbak et al. (2022). This interpretation also highlights a close connection to well-established TTA approaches used in generative models (Korbak et al., 2022; Mudgal et al., 2024; Li et al., 2024), where the model is viewed as an RL policy fine-tuned with a reward function encoding evidence or alignment criteria. In this view, EPHAD performs a KL-regularised shift of the contaminated density $f_X^\pm$ toward regions favored by the evidence function $T(x)$ while preserving consistency with $f_X^\pm$ through the KL term.

## 4.1 Extension to score-based anomaly detection

Since estimating explicit densities is often infeasible in high-dimensional spaces, most modern AD methods rely on *scores* rather than PDFs. Recall that the inlier score function is an order-preserving transformation of the inlier PDF, i.e., $s_{\text{in}}^+(x) = \phi(f_X^+(x))$, where $\phi(\cdot)$ is a monotonic transformation such as the logarithm. When trained on contaminated data $\mathcal{D}_{\text{train}}^\pm$, the model learns a contaminated inlier score $s_{\text{in}}^\pm(x) = \phi(f_X^\pm(x))$. Although $\phi$ is typically unknown and possibly non-invertible, the sample ranking induced by $s_{\text{in}}^\pm(x)$ is identical to that induced by $f_X^\pm(x)$.

Following energy-based model (EBM) formulations (LeCun et al., 2006), we can define the associated contaminated PDF as

$$\tilde{f}_X^\pm(x) = \frac{\exp(s_{\text{in}}^\pm(x))}{Z_X^e}, \tag{6}$$

where $Z_X^e = \int_X \exp(s_{\text{in}}^\pm(x))$ is the normalising constant. Applying exponential tilting to $\tilde{f}_X^\pm(x)$ as in (3), we obtain:

$$\check{\tilde{f}}_X^\pm(x) = \frac{\tilde{f}_X^\pm(x) \exp(T(x)/\beta)}{Z_X^\beta} = \frac{\exp(s_{\text{in}}^\pm(x)) \exp(T(x)/\beta)}{Z_X^\beta Z_X^e}. \tag{7}$$

Under Proposition 4.1, when condition (4) holds, the revised $\check{\tilde{f}}_X^\pm$ is closer to the true inlier density $f_X^+$ in KL divergence than the unadjusted $\tilde{f}_X^\pm$. Because AD depends only on the relative ordering of samples, the normalization constants in (7) can be ignored. The exponential mapping is strictly monotonic, so ranking and decision regions are preserved. Consequently, we can write

$$\check{\tilde{f}}_X^\pm(x) \propto \exp(s_{\text{in}}^\pm(x) + T(x)/\beta) := \check{s}_{\text{in}}^\pm(x), \tag{8}$$

where we define $\check{s}_{\text{in}}^\pm$ as the revised inlier score. The anomaly detector in (1) can thus be redefined as

$$g_{\lambda_s}(x) = \begin{cases} +1, & \text{if } \check{s}_{\text{in}}^\pm(x) \geq \lambda_s, \\ -1, & \text{otherwise.} \end{cases} \tag{9}$$

This extension allows EPHAD to operate directly on score-based AD models, enabling post-hoc correction of models trained on contaminated datasets without requiring retraining or access to the original training procedure. In all subsequent experiments, we adopt this score-based formulation of EPHAD, reflecting the dominance of score-based methods in modern anomaly detection practice.

## 4.2 An illustrative example

To illustrate the effect of EPHAD, we use a toy dataset inspired by Qiu et al. (2022). The dataset is generated using a two-dimensional mixture model comprising three Gaussian components: $c_1 := \mathcal{N}(\mu_1, \Sigma_1), c_2 := \mathcal{N}(\mu_2, \Sigma_2), c_3 := \mathcal{N}(\mu_3, \Sigma_3)$. Here, each component follows a Gaussian distribution $\mathcal{N}(\mu, \Sigma)$ with mean $\mu$ and covariance $\Sigma$. Normal samples are drawn from $f_X^+ = c_1$,

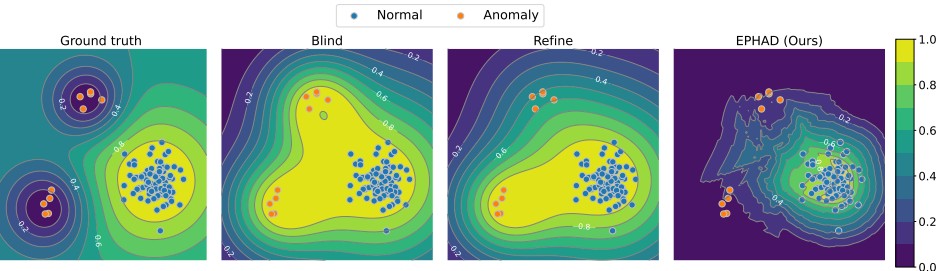

Figure 1: DeepSVDD trained on 2D synthetic contaminated training data with different configurations: (I) Supervised AD with ground truth labels for reference, (ii)"Blind" considering all samples as normal, (iii) "Refine" filtering out a fraction of the anomalies, and (iv) EPHAD updating the "Blind" anomaly detector using evidence computed on the samples available at test-time.

with $\mu_1 = [1,1]^T$ and $\Sigma_1 = 0.07\,\mathbf{I}_2$. Anomalous samples are drawn from a mixture distribution $f_X^- := 0.5c_2 + 0.5c_3$ where $\mu_2 = [-0.25, 2.5]^T, \mu_3 = [-1, 0.5]^T$ and $\Sigma_2 = \Sigma_3 = 0.03\,\mathbf{I}_2$. The extended implementation details is provided in Appendix B.2. Using this setting, we create a contaminated dataset consisting 100 data points. We compare the baseline DeepSVDD (Ruff et al., 2018) across three configurations as illustrated in Figure 1: (i) "Blind", (ii) "Refine", and (iii) with EPHAD. We refer to the baseline model that treats all samples as normal as "Blind", while "Refine" denotes a model that iteratively filters out suspected anomalies during training. As an evidence function in EPHAD, LOF (Breunig et al., 2000) is computed on test samples at test time. The results in Figure 1 demonstrate that the "Blind" configuration mistakenly considers all anomalies as normal. The "Refine" configuration improves performance by filtering out a subset of anomalies. Finally, EPHAD establishes a clearer boundary around normal samples.

## 4.3 Determining the temperature parameter $\beta$

As previously discussed, EPHAD has only a single hyperparameter, $\beta$, which controls the trade-off between reliance on the original AD model and the evidence function $T(x)$. A straightforward approach to selecting $\beta$ would involve evaluating the AD performance of the prior and $T(x)$ individually on a validation set and choosing $\beta$ accordingly. However, this strategy introduces additional computational overhead at test time and requires access to a labelled validation set of sufficient size to ensure reliable performance estimation – conditions often impractical in real-world deployments. To address this limitation, we propose an adaptive extension of our approach, termed EPHAD-Ada, that determines the optimal $\beta$ in an unsupervised manner using only test data at test time. This adaptation is inspired by the principle of Entropy Minimisation (EM) (Press et al., 2024), a widely-used technique in test-time adaptation (Xiao and Snoek, 2024). Motivated by the observation from Wang et al. (2021) that models tend to be more accurate when predictions are made with high confidence, we apply it to compute the hyperparameter $\beta$. Specifically, the computation of $\beta$ depends on the entropy of the inlier probabilities derived from the scores of both the original model and the evidence function.

**Computing inlier probability from the output scores**. For an output score $s \in \mathbb{R}$, the class label given the score can be modelled as a conditional random variable $Y \mid S = s$. Following this, the inlier probability can be expressed as

$$p_{Y=+1}(s) := \mathbb{P}(Y = +1 \mid S = s) = \mathbb{P}(S > s) = 1 - p_s, \tag{10}$$

where $p_s := \mathbb{P}(S \le s)$. Since $p_s$ is unknown in practice, we follow the approach of Perini et al. (2021) and treat it as a random variable $P_s$ with a prior distribution $\text{Beta}(1, 1)$, corresponding to a uniform prior over $[0, 1]$. Given that the label $Y \in \{+1, -1\}$, we model the conditional distribution $Y \mid S = s$ as a Bernoulli random variable. To estimate $p_s$, we draw samples $s' \sim S$ by first sampling $x \sim \mathcal{X}$ and then computing the corresponding anomaly score $s'$. We record a success ($b = 1$) if $s' \le s$, and a failure ($b = 0$) otherwise. Repeating this procedure $n$ times yields $t$ successes and $n - t$ failures. Then, according to Theorem 2 in Perini et al. (2021), the posterior distribution of $P_s$ given the observed binary outcomes $b_1, \ldots, b_n$ is $\text{Beta}(1 + t, 1 + n - t)$. We estimate $p_s$ using the posterior mean of $P_s$ as

$$p_s := \mathbb{E}[P_s] = \frac{1 + t}{2 + n}. \tag{11}$$

In practice, the posterior is inferred from test samples, so the sample size $n$ is constrained by the number of available test points. Finally, combining Equations (10) and (11), we obtain the estimated inlier probability for a data point $x$ with anomaly score $s$ as

$$p_{Y=+1}(s) = 1 - p_s = 1 - \frac{1+t}{2+n}. \tag{12}$$

Finally, using (12), we compute the inlier probabilities $p_{Y=+1}^o(x) := p_{Y=+1}(s_{\text{in}}^{\pm}(x))$ and $p_{Y=+1}^e(x) := p_{Y=+1}(T(x))$ from the scores of the original model and the evidence function, respectively.

**Computing the value of the hyperparameter $\beta$.** We define the empirical entropy of the binary predictive PMF $p_Y(x)$ as

$$H(p_Y) = - \sum_{x \in \mathcal{D}_{\text{test}}} \left[ p_{Y=+1}(x) \log p_{Y=+1}(x) + p_{Y=-1}(x) \log p_{Y=-1}(x) \right]. \tag{13}$$

The adaptive temperature parameter is then defined as

$$\beta_{\text{ada}} = \frac{H(p_Y^e)}{H(p_Y^o) + \delta}, \tag{14}$$

where $\delta > 0$ is a small constant introduced to ensure numerical stability. A low $H(p_Y^o)$ indicates that the original AD model produces confident (low-entropy) predictions, suggesting that a higher value of $\beta$ should be used to place greater trust in this model. Conversely, a lower $H(p_Y^e)$ implies higher confidence in the evidence function, motivating a smaller $\beta$. Through this formulation, `EPHAD-Ada` enables unsupervised, test-time determination of $\beta$, thereby improving practicality and eliminating the need for labelled validation data.

## 5 Experiments

We evaluate the effectiveness of `EPHAD` for unsupervised AD across a range of datasets, including visual AD datasets (Section 5.1), tabular AD datasets (Section 5.2), and an industrial AD use case (Appendix C.2). To systematically investigate the impact of contamination at different levels in a rigorous and reproducible way, we introduce controlled contamination into the data, adhering to the experimental design employed in several prior studies (Jiang et al., 2022; Wang et al., 2025; Zhou and Wu, 2024). The evidence functions employed in the experiments are computed in an unsupervised manner without utilising ground-truth labels in the test set $\mathcal{D}_{\text{test}}$, mitigating the risk of overfitting. Unless stated otherwise, we use a contamination factor of $\epsilon = 0.1$ and a parameter $\beta = 0.5$. An ablation study on different values of $\epsilon$ and $\beta$ is presented in Section 5.3. For image and tabular datasets, we evaluate performance using the AUROC. Following prior work (Roth et al., 2022; Gudovskiy et al., 2022), AUROC is averaged across all categories for each dataset.

### 5.1 Experiments on visual AD datasets

**Benchmark datasets**. We assess the effectiveness of `EPHAD` in both sensory and semantic anomaly detection. Sensory AD focuses on detecting physical defects or imperfections, such as a broken capsule or a cut in a carpet, while semantic AD identifies anomalies belonging to a different semantic class—for instance, treating cats as normal and any other animal as anomalous. For sensory AD in industrial contexts, we evaluate performance using four well-established benchmark datasets: MVTecAD (Bergmann et al., 2019), MPDD (Jezek et al., 2021), ViSA (Zou et al., 2022), and RealIAD (Wang et al., 2024). For semantic AD, we utilise four commonly used datasets, including CIFAR-10, Fashion-MNIST, MNIST, and SVHN. Following the one-vs-rest protocol (Qiu et al., 2022), we construct $k$ AD tasks per dataset, where $k$ corresponds to the number of classes. For MVTecAD, ViSA, MPDD and RealIAD, we adopt the "overlap" setting, introducing $\epsilon\%$ contamination into the training set by randomly selecting anomalous samples from the test set while retaining them in the test set Jiang et al. (2022). For the remaining datasets, we follow the "non-overlapping" setting, excluding anomalous samples used for contamination simulation from the test set. Our implementation is based on the public codebase from Jiang et al. (2022). Additional details are provided in Appendix B.1.

**Baseline AD methods**. We evaluate the performance of several state-of-the-art unsupervised anomaly detection methods, including PatchCore (Roth et al., 2022), PaDim (Defard et al., 2021), CFLOW

Table 1: Performance on both sensory and semantic AD benchmarking datasets with $10\%$ contamination ratio. Style: AUROC % ($\pm$ SE). Best in **bold**.

| Method | Non-overlap | | | | | Overlap | | |
|---|---|---|---|---|---|---|---|---|
| | MNIST | FMNIST | CIFAR10 | SVHN | RealIAD | MVTec | MPDD | ViSA |
| CLIP | 71.15 | 95.63 | 98.63 | 58.46 | 65.74 | 86.34 | 60.02 | 74.47 |
| CFLOW | 77.24 (± 1.01) | 72.87 (± 0.48) | 65.47 (± 0.02) | 55.09 (± 0.09) | **76.42** (± 0.47) | 87.58 (± 0.77) | 66.69 (± 2.06) | 75.71 (± 1.28) |
| + EPHAD | **78.40** (± 0.81) | **92.97** (± 0.19) | **97.38** (± 0.01) | **55.82** (± 0.06) | 71.58 (± 0.17) | 87.98 (± 0.12) | 65.22 (± 0.93) | 78.53 (± 0.27) |
| + EPHAD-Ada | 78.08 (± 0.91) | 91.63 (± 0.29) | 96.43 (± 0.0) | 55.78 (± 0.04) | 73.86 (± 0.24) | **89.84** (± 0.3) | **67.81** (± 1.63) | **79.64** (± 0.63) |
| DRÆM | 71.44 (± 0.29) | 76.53 (± 0.18) | 63.41 (± 0.26) | 51.55 (± 0.07) | 67.46 (± 0.21) | 70.55 (± 1.97) | 62.32 (± 1.96) | 69.61 (± 1.57) |
| + EPHAD | **73.51** (± 0.39) | **92.46** (± 0.25) | **97.17** (± 0.02) | **54.18** (± 0.07) | 69.89 (± 0.23) | 87.13 (± 0.39) | 67.02 (± 0.29) | **76.89** (± 0.99) |
| + EPHAD-Ada | 72.88 (± 0.33) | 84.96 (± 0.97) | 87.73 (± 1.52) | 53.79 (± 0.36) | **70.15** (± 0.05) | **87.24** (± 0.39) | **69.55** (± 0.42) | 74.95 (± 1.15) |
| FastFlow | 82.65 (± 0.43) | 83.66 (± 0.06) | 62.94 (± 0.37) | 54.02 (± 0.11) | **82.03** (± 0.08) | 84.24 (± 1.07) | **71.94** (± 0.87) | 77.83 (± 0.22) |
| + EPHAD | **83.20** (± 0.43) | **93.49** (± 0.07) | **97.34** (± 0.02) | 55.07 (± 0.07) | 77.22 (± 0.08) | 87.68 (± 0.5) | 66.84 (± 0.34) | 80.29 (± 0.07) |
| + EPHAD-Ada | 82.83 (± 0.44) | 92.10 (± 0.14) | 96.24 (± 0.05) | **55.26** (± 0.17) | 81.1 (± 0.06) | **88.07** (± 0.8) | 70.08 (± 0.41) | **80.71** (± 0.08) |
| PaDiM | 87.50 (± 0.23) | 86.84 (± 0.06) | 62.53 (± 0.4) | 55.49 (± 0.28) | **80.39** (± 0.35) | 77.85 (± 0.43) | 36.58 (± 2.58) | 73.07 (± 0.27) |
| + EPHAD | 87.45 (± 0.22) | **94.66** (± 0.03) | **97.10** (± 0.03) | 56.94 (± 0.22) | 75.94 (± 0.25) | **86.58** (± 0.38) | **55.48** (± 0.72) | **77.73** (± 0.27) |
| + EPHAD-Ada | **87.56** (± 0.23) | 92.87 (± 0.02) | 90.23 (± 0.67) | **57.09** (± 1.05) | 79.56 (± 0.28) | 86.10 (± 0.52) | 49.06 (± 1.52) | 76.62 (± 0.38) |
| PatchCore | 86.33 (± 0.09) | 78.97 (± 0.06) | 75.69 (± 0.09) | **69.64** (± 0.04) | 70.08 (± 0.07) | 70.51 (± 0.7) | 53.58 (± 0.54) | 27.2 (± 0.31) |
| + EPHAD | 86.36 (± 0.1) | **94.73** (± 0.01) | **97.74** (± 0.01) | 61.31 (± 0.0) | 69.76 (± 0.2) | **86.45** (± 0.14) | **60.58** (± 1.12) | **62.94** (± 0.41) |
| + EPHAD-Ada | **86.38** (± 0.1) | 89.99 (± 0.2) | 96.63 (± 0.09) | 68.4 (± 0.52) | **77.18** (± 0.09) | 83.53 (± 0.18) | 56.97 (± 1.23) | 48.60 (± 0.51) |
| RD | 77.33 (± 0.09) | 84.11 (± 0.72) | 66.29 (± 0.31) | 55.54 (± 0.58) | **89.13** (± 0.18) | 80.08 (± 1.32) | **75.08** (± 1.75) | **86.33** (± 0.46) |
| + EPHAD | 78.19 (± 0.28) | **95.77** (± 0.03) | **98.40** (± 0.0) | 57.38 (± 0.14) | 69.35 (± 0.26) | 85.82 (± 0.31) | 62.62 (± 0.27) | 77.76 (± 0.19) |
| + EPHAD-Ada | **78.91** (± 0.21) | 95.64 (± 0.04) | 98.0 (± 0.17) | 57.78 (± 0.5) | 72.78 (± 0.43) | **86.69** (± 0.38) | 63.97 (± 0.88) | 79.42 (± 0.34) |
| ULSAD | **90.83** (± 0.08) | 88.64 (± 0.13) | 72.45 (± 0.18) | **64.27** (± 0.22) | **89.06** (± 0.01) | 91.93 (± 0.15) | **77.67** (± 0.42) | 86.58 (± 0.13) |
| + EPHAD | 90.41 (± 0.06) | **95.03** (± 0.07) | **97.90** (± 0.02) | 58.17 (± 0.18) | 80.58 (± 0.06) | 91.31 (± 0.06) | 72.79 (± 1.05) | 85.82 (± 0.1) |
| + EPHAD-Ada | 90.8 (± 0.07) | 94.55 (± 0.08) | 97.29 (± 0.02) | 59.68 (± 0.16) | 85.84 (± 0.04) | **92.25** (± 0.07) | 76.31 (± 1.04) | **87.23** (± 0.05) |

(Gudovskiy et al., 2022), FastFLOW (Yu et al., 2021), DRÆM (Zavrtanik et al., 2021), Reverse Distillation (RD) (Deng and Li, 2022), and ULSAD (Patra and Ben Taieb, 2024), both with and without the integration of EPHAD. Implementations for all methods, except ULSAD, are based on the Anomalib library (Akcay et al., 2022), while ULSAD is implemented using its official public code. Since, to the best of our knowledge, no existing AD method with contaminated data offers post-hoc adaptation in the same manner as EPHAD, our primary objective is to demonstrate the effectiveness of EPHAD by comparing its relative performance against the AD model and the evidence function alone. We also provide comparative analyses with three existing frameworks "Refine" (Yoon et al., 2022), Latent Outlier Exposure (LOE) (Qiu et al., 2022), and SoftPatch (Jiang et al., 2022) in Appendix C.3.

**Evidence function**. For the experiments, we employ Contrastive Language-Image Pre-training (CLIP) (Radford et al., 2021) as the evidence function for image-based datasets, following the anomaly detection approach as in WinCLIP (Jeong et al., 2023). We use CLIP as the evidence function $T(x)$ in EPHAD. We start by defining two lists of textual prompt templates, $\mathcal{T}_N = \{n_1, \cdots, n_k\}$ and $\mathcal{T}_A = \{a_1, \cdots, a_k\}$, corresponding to normal and anomalous classes, respectively. These templates are dataset-dependent, reflecting subjectivity (e.g., "missing wire" as anomalous for cables). For each label, compute the mean of text embeddings $t_N$ and $t_A$. Finally, given an input image $x$, the evidence $T(x)$ at test-time is computed as:

$$T(x) := \frac{\exp\left(\langle e_i(x), t_N \rangle / \gamma\right)}{\exp\left(\langle e_i(x), t_N \rangle / \gamma\right) + \exp\left(\langle e_i(x), t_A \rangle / \gamma\right)}.$$

Additional implementation details are provided in Appendix B.3.1.

While CLIP has been previously applied as a standalone zero-shot anomaly detector, our methodology leverages it differently: we employ CLIP not as a complete detection system, but as an auxiliary source of evidence integrated into a more general and flexible framework. Importantly, EPHAD is not limited to foundation models such as CLIP; it can seamlessly incorporate domain-specific knowledge as well (see Section C.2), thereby broadening its applicability across diverse domains.

**Results**. In our experiments, as we adopt CLIP in the same manner as WinCLIP (Jeong et al., 2023), the baseline CLIP results reported here directly correspond to the standalone performance of WinCLIP. In Table 1, we observe that while zero-shot AD using CLIP performs well on real-world image datasets such as CIFAR10 and FMNIST, its effectiveness declines on domain-specific datasets like MVTec, MPDD, and ViSA, where existing AD methods, such as ULSAD, achieve superior performance. However, when these AD methods are used within the EPHAD framework with CLIP as an evidence function in a post-hoc manner, their performance improves in most cases. Notably, even when CLIP-based AD alone does not achieve the best results, as seen in SVHN, incorporating it

Table 2: Performance on tabular AD benchmarking datasets with $10\%$ contamination ratio. Style: AUROC % ($\pm$ SE). Best in **bold**. † represents transductive inference.

| Dataset | aloi | cover | glass | ionosphere | letter | pendigits | vowels | wine |
|---|---|---|---|---|---|---|---|---|
| LOF† | 72.64 ($\pm$ 0.1) | 52.12 ($\pm$ 0.1) | 77.52 ($\pm$ 0.93) | 82.43 ($\pm$ 0.16) | 83.15 ($\pm$ 0.73) | 47.21 ($\pm$ 0.12) | 89.1 ($\pm$ 0.67) | 97.57 ($\pm$ 1.46) |
| COPOD | 51.46 ($\pm$ 0.05) | 78.7 ($\pm$ 0.03) | 76.11 ($\pm$ 0.77) | 79.42 ($\pm$ 1.03) | 56.71 ($\pm$ 0.12) | **88.44** ($\pm$ 0.2) | 56.1 ($\pm$ 0.32) | 80.51 ($\pm$ 1.36) |
| + EPHAD | 52.55 ($\pm$ 0.06) | 79.01 ($\pm$ 0.02) | 79.45 ($\pm$ 0.95) | 81.67 ($\pm$ 0.95) | 57.62 ($\pm$ 0.09) | 88.38 ($\pm$ 0.2) | 58.87 ($\pm$ 0.34) | 86.78 ($\pm$ 1.96) |
| + EPHAD-Ada | **53.65** ($\pm$ 0.17) | **79.57** ($\pm$ 0.01) | **81.77** ($\pm$ 1.28) | **84.15** ($\pm$ 0.38) | **71.03** ($\pm$ 0.99) | 87.09 ($\pm$ 0.22) | **75.39** ($\pm$ 0.88) | **93.96** ($\pm$ 1.66) |
| DeepSVDD | 54.06 ($\pm$ 0.54) | 75.11 ($\pm$ 11.37) | 64.52 ($\pm$ 6.87) | 83.09 ($\pm$ 0.57) | 50.51 ($\pm$ 2.54) | **74.87** ($\pm$ 9.91) | 64.47 ($\pm$ 2.55) | 82.26 ($\pm$ 2.29) |
| + EPHAD | 64.36 ($\pm$ 0.21) | **75.74** ($\pm$ 11.06) | 80.94 ($\pm$ 3.31) | 84.9 ($\pm$ 0.17) | 61.26 ($\pm$ 2.42) | 72.68 ($\pm$ 8.72) | 76.61 ($\pm$ 1.24) | 92.94 ($\pm$ 1.74) |
| + EPHAD-Ada | **70.67** ($\pm$ 0.22) | 75.58 ($\pm$ 10.82) | 80.94 ($\pm$ 2.52) | **85.03** ($\pm$ 0.25) | 65.9 ($\pm$ 2.88) | 74.08 ($\pm$ 9.18) | **82.12** ($\pm$ 0.9) | **93.96** ($\pm$ 1.77) |
| ECOD | 53.14 ($\pm$ 0.03) | 85.34 ($\pm$ 0.02) | 67.65 ($\pm$ 0.44) | 73.04 ($\pm$ 0.84) | 56.41 ($\pm$ 0.29) | 90.63 ($\pm$ 0.17) | 54.29 ($\pm$ 0.06) | 67.12 ($\pm$ 2.04) |
| + EPHAD | 54.33 ($\pm$ 0.05) | **85.45** ($\pm$ 0.02) | 72.59 ($\pm$ 0.61) | 74.34 ($\pm$ 0.85) | 57.17 ($\pm$ 0.29) | **90.65** ($\pm$ 0.17) | 56.82 ($\pm$ 0.14) | 74.97 ($\pm$ 2.88) |
| + EPHAD-Ada | **55.47** ($\pm$ 0.18) | **85.45** ($\pm$ 0.01) | 78.43 ($\pm$ 1.72) | **78.14** ($\pm$ 0.49) | 70.15 ($\pm$ 1.15) | 89.66 ($\pm$ 0.2) | **75.39** ($\pm$ 0.91) | 89.27 ($\pm$ 2.95) |
| IForest | 54.05 ($\pm$ 0.21) | 72.59 ($\pm$ 1.59) | 78.5 ($\pm$ 1.47) | 89.58 ($\pm$ 1.57) | 59.84 ($\pm$ 0.64) | **81.86** ($\pm$ 1.48) | 66.01 ($\pm$ 0.57) | 80.4 ($\pm$ 3.42) |
| + EPHAD | **71.75** ($\pm$ 0.08) | 63.64 ($\pm$ 0.92) | 79.12 ($\pm$ 1.01) | 83.5 ($\pm$ 0.16) | **81.53** ($\pm$ 0.59) | 55.56 ($\pm$ 0.98) | **88.59** ($\pm$ 0.65) | **97.51** ($\pm$ 1.51) |
| + EPHAD-Ada | 57.49 ($\pm$ 0.31) | **73.15** ($\pm$ 1.57) | **83.15** ($\pm$ 1.86) | **90.05** ($\pm$ 1.22) | 71.38 ($\pm$ 0.86) | 79.5 ($\pm$ 1.5) | 80.76 ($\pm$ 0.6) | 93.56 ($\pm$ 2.15) |
| LOF | 73.57 ($\pm$ 0.1) | 22.44 ($\pm$ 0.1) | 71.79 ($\pm$ 1.08) | **94.64** ($\pm$ 0.52) | **85.74** ($\pm$ 0.54) | 14.87 ($\pm$ 0.18) | **93.04** ($\pm$ 0.54) | **99.94** ($\pm$ 0.05) |
| + EPHAD | 73.62 ($\pm$ 0.07) | **44.2** ($\pm$ 0.07) | **76.4** ($\pm$ 0.68) | 89.74 ($\pm$ 0.55) | 84.84 ($\pm$ 0.39) | **37.64** ($\pm$ 0.13) | 91.3 ($\pm$ 0.1) | **99.94** ($\pm$ 0.05) |
| + EPHAD-Ada | **73.85** ($\pm$ 0.05) | 36.78 ($\pm$ 0.23) | 75.67 ($\pm$ 0.75) | 91.85 ($\pm$ 0.68) | 85.31 ($\pm$ 0.36) | 30.16 ($\pm$ 1.01) | 91.85 ($\pm$ 0.12) | **99.94** ($\pm$ 0.05) |

within EPHAD still leads to significant improvements. For instance, CFLOW, PaDiM, and RD exhibit enhanced performance after using EPHAD, surpassing both CLIP and the standalone AD methods. This highlights the effectiveness of EPHAD in refining anomaly scores for better AD performance. In some cases, such as ULSAD on SVHN, we observe a decline in performance when integrating EPHAD compared to the standalone AD method. This typically occurs when the AD method substantially outperforms the evidence function. In such scenarios, overly relying on the evidence can diminish overall performance. To mitigate this effect, careful tuning of $\beta$ enables the framework to adapt effectively to different datasets, AD methods, and evidence functions. A detailed analysis of the impact of varying $\beta$ values is presented in Section 5.3.

Using the adaptive variant, EPHAD-Ada, we observe further improvements in certain settings, such as with PatchCore and DREAM on the RealIAD dataset. Interestingly, in cases where the default value of $\beta = 0.5$ led to decreased performance (e.g., ULSAD on SVHN or MPDD), EPHAD-Ada manages to overcome the problem, highlighting its effectiveness. Nevertheless, while EPHAD-Ada offers an unsupervised mechanism for determining $\beta$, its performance is often comparable to, or slightly below, that of EPHAD with the default value for $\beta$.

## 5.2 Experiments on tabular AD datasets

**Benchmark datasets**. We evaluate our proposed framework on 26 classical benchmark datasets from ADBench (Han et al., 2022). The classical datasets include datasets from different domains such as healthcare (e.g., annthyroid, breastw), astronautics (e.g. satellite), and finance (fraud). Following Qiu et al. (2022), we preprocess, split the dataset into the train and test sets and simulate contamination using synthetic anomalies created by adding zero-mean Gaussian noise with a large variance to the anomalous sample from the test set.

**Baseline AD methods**. We compare EPHAD against IFOREST (Liu et al., 2012), LOF (Breunig et al., 2000), DeepSVDD (Ruff et al., 2018), ECOD (Li et al., 2023b) and COPOD (Li et al., 2020) using ADBench (Han et al., 2022).

**Evidence function**. We use the output of Local Outlier Factor (LOF) (Breunig et al., 2000) and Isolation Forest (IForest) (Liu et al., 2012). Additional details provided in the Appendix B.3.2.

**Results**. The experimental results for a subset of the 26 benchmarking datasets are presented in Table 2, with the extended version provided in Appendix C.1. We observe that most AD methods benefit from our post-hoc adjustment framework EPHAD, often achieving performance improvements that surpass both the evidence function and the AD method in isolation. For example, COPOD, when updated with LOF as the evidence function on cover, glass and pendigits datasets, shows this behaviour. Additionally, as seen in the image-based experiments, performance degradation in certain cases arises when the framework places excessive emphasis on an evidence function that is substantially weaker than the AD method. However, as previously discussed, this limitation can be mitigated by appropriately tuning $\beta$. Similar to the results in the image-based experiments, we

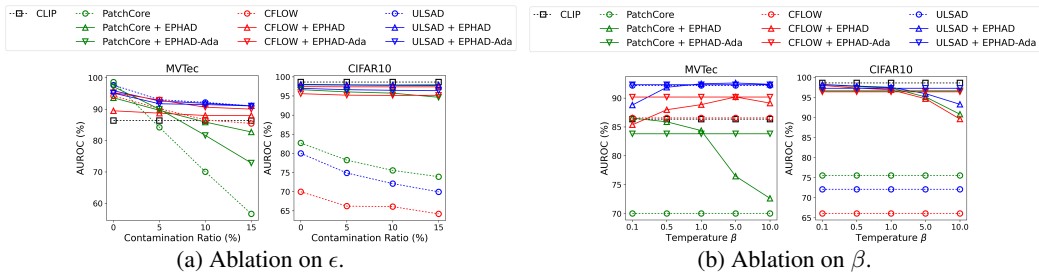

(a) Ablation on $\epsilon$.  (b) Ablation on $\beta$.

Figure 2: Ablation on parameters.

observe improvements when using the adaptive variant `EPHAD-Ada`. In some scenarios, we also observe that `EPHAD-Ada` avoids the performance drop observed with `EPHAD`, such as with LOF on the ionosphere dataset and with DeepSVDD on the pendigits dataset. Nonetheless, the performance in most cases is similar to `EPHAD` with default value $\beta$, suggesting the need for further exploration.

### 5.3 Ablation study

In this section, we first analyse the sensitivity of `EPHAD` to various contamination ratios. Then, we investigate the effect of the temperature $\beta$ on AD performance.

**Effect of varying contamination ratio**. Here, we evaluate the sensitivity of our proposed framework by varying the contamination ratio $\{0\%, 5\%, 10\%, 15\%\}$. The results are summarised in the Figure 2a. Applying `EPHAD` results in improvements across all contamination ratios for most of the AD methods. Furthermore, in the presence of a strong evidence function, such as CLIP, we can observe that the performance becomes almost constant even as the contamination ratio increases from $5\%$ to $15\%$. An extended version is provided in Figure 3.

**Effect of temperature parameter** $\beta$. We also analyse the performance of the `EPHAD` by varying the temperature parameter $\beta$. In Figure 2b, we can see how $\beta$ allows for controlling the trade-off between the prior AD method and the evidence. As discussed earlier, we observe that setting $\beta \approx 0$ results in full reliance on $T(x)$, while with increasing $\beta$, $T(x)$ is disregarded and it defaults to the prior. Additionally, `EPHAD-Ada` achieves performance comparable to the best configuration of `EPHAD` across the explored range of $\beta$, highlighting its effectiveness. An extended version is provided in Figure 4.

## 6 Conclusion

**Limitations and future work**. While existing AD methods can serve as domain-agnostic evidence functions within `EPHAD`, the full potential of our framework is best realised by designing evidence functions that incorporate domain-specific knowledge. Exploring the interplay between datasets, AD methods, and evidence functions remains an open direction for future work. Another limitation concerns the parameter $\beta$, which has a significant influence on overall performance, as demonstrated in our experiments. Although we introduced an unsupervised strategy for estimating $\beta$ in `EPHAD-Ada`, this approach does not always lead to performance improvements. We hypothesize that this may stem from uncalibrated inlier probability. Future work should thus investigate more reliable approaches for inferring $\beta$ based on the anomaly scores and the underlying distributions of normal and anomalous samples in the test set. Finally, integrating explainability techniques into `EPHAD` represents an interesting direction for future research, as it could provide deeper insights for real-world applications.

**Concluding remarks**. Unsupervised AD methods typically assume anomaly-free training data, yet real-world datasets often contain undetected or mislabeled anomalies, leading to significant performance degradation. Existing approaches to address contamination often require access to model parameters, training data, or the training pipeline, limiting their practicality in real-world deployments. In this work, we introduce `EPHAD`, a simple, post-hoc adjustment framework that refines the outputs of any AD method trained on contaminated data by incorporating evidence collected at test-time. Extensive experiments demonstrate the effectiveness of `EPHAD` across diverse sources of evidence, multiple AD methods, and various datasets. Additionally, ablation studies analyse the impact of hyperparameters and varying contamination levels, highlighting the robustness of `EPHAD`.

## Acknowledgments and Disclosure of Funding

This work is supported by the FLARACC research project (Federated Learning and Augmented Reality for Advanced Control Centers), funded by the Wallonia region in Belgium.

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

# A Proofs

## A.1 Proof of Proposition 4.1

*Proof.* From (2), we have

$$f_X^{\pm}(x) = \epsilon f_X^{-}(x) + (1 - \epsilon) f_X^{+}(x).$$

Additionally, from (3), we have

$$\check{f}_X^{\pm}(x) = \frac{f_X^{\pm}(x) \exp(T(x)/\beta)}{Z_X^{\beta}}$$

Then,

$$
\begin{aligned}
D_{\mathrm{KL}}(f_X^+ \| \check{f}_X^{\pm}) &= \mathbb{E}_{x \sim \mathrm{P}_X^+} \left[ \log \frac{f_X^+(x)}{\check{f}_X^{\pm}(x)} \right] \\
&= \mathbb{E}_{x \sim \mathrm{P}_X^+} \left[ \log f_X^+(x) - \log \check{f}_X^{\pm}(x) \right] \\
&= \mathbb{E}_{x \sim \mathrm{P}_X^+} \left[ \log f_X^+(x) - \log \frac{f_X^{\pm}(x) \exp(T(x)/\beta)}{Z_X^{\beta}} \right] \\
&= \mathbb{E}_{x \sim \mathrm{P}_X^+} \left[ \log f_X^+(x) - \log f_X^{\pm}(x) - \frac{T(x)}{\beta} + \log Z_X^{\beta} \right] \\
&= D_{\mathrm{KL}}(f_X^+ \| f_X^{\pm}) - \mathbb{E}_{x \sim \mathrm{P}_X^+} \left[ \frac{T(x)}{\beta} - \log Z_X^{\beta} \right] \\
&= D_{\mathrm{KL}}(f_X^+ \| f_X^{\pm}) - \mathbb{E}_{x \sim \mathrm{P}_X^+} \left[ \log \frac{\exp(T(x)/\beta)}{Z_X^{\beta}} \right].
\end{aligned}
$$

We aim to increase the alignment between $f_X^+$ and $\check{f}_X^{\pm}$. Since the KL divergence is non-negative, if the expectation term is positive, we obtain

$$D_{\mathrm{KL}}(f_X^+ \| \check{f}_X^{\pm}) \le D_{\mathrm{KL}}(f_X^+ \| f_X^{\pm}).$$

Therefore, the following condition should hold:

$$\mathbb{E}_{x \sim \mathrm{P}_X^+} \left[ \log \frac{\exp(T(x)/\beta)}{Z_X^{\beta}} \right] \ge 0.$$

$\square$

# B Additional implementation details

## B.1 Benchmark datasets

For sensory AD in industrial settings, we use three widely recognised benchmark datasets. MVTecAD (Bergmann et al., 2019) comprises images from 15 categories (10 objects and 5 textures) with 3629 normal training images and 1258 anomalous and 467 normal test images, each containing pixel-level annotations of defects. MPDD (Jezek et al., 2021) targets metal part defects under varying conditions, offering 888 training images and test datasets consisting of 176 normal and 282 anomalous images across 6 metal part categories. ViSA (Zou et al., 2022) provides 10821 high-resolution images (9621 normal and 1200 anomalous) spanning 12 categories, capturing a range of anomalies such as scratches, cracks, missing parts, and misplacements. Each defect type is represented by 15–20 images, and some images feature multiple defects. RealIAD (Wang et al., 2024) is a large-scale industrial AD dataset comprising $\sim 150k$ images across 30 categories and having various types of defects such as scratches, dirt and missing parts. For experiments with RealIAD, we use the training split with $10\%$ contamination and the test split provided by the authors. For the semantic datasets, using the one-vs-rest protocol, we create $k$ AD tasks for each dataset, where $k$ is the number of classes. In each task, one class is designated as normal, while the remaining classes are treated as anomalous. Across both sensory and semantic AD, the training datasets consist of a mixture of normal samples and a fraction $\epsilon$ of anomalous samples, reflecting realistic contamination scenarios.

## B.2 Details of the experiment using synthetic Data

The synthetic dataset is generated using a 2D Gaussian mixture model with three components. Normal samples are drawn from $f_X^+(x) := \mathcal{N}([1,1]^T, 0.07\mathbf{I}_2)$, while anomalous samples are sampled from $f_X^-(x) := \mathcal{N}([-0.25, 2.5]^T, 0.03\mathbf{I}_2) + \mathcal{N}([-1, 0.5]^T, 0.03\mathbf{I}_2)$. For the experiments, we use DeepSVDD with a one-layer radial basis function (RBF) network. The hidden layer comprises three neurons, with their centres fixed at the mean of each Gaussian component, while the scales are optimized during training. The RBF network outputs a 1D scalar obtained as a linear combination of the outputs from the hidden layer. The centre is initialized randomly and made trainable, with an added bias term in the final layer. Although these modifications are not recommended by Ruff et al. (2018) to avoid collapse to a trivial solution, Qiu et al. (2022) observed that these changes enhance model flexibility and convergence. Following this, we train DeepSVDD using the Adam optimizer with a learning rate of 0.01, 200 epochs, and a mini-batch size of 25.

## B.3 Computing evidence functions

EPHAD relies on an evidence function $T(x)$, computed during test-time, to refine anomaly scores by assigning higher values to samples from $\mathrm{P}_X^+$ than those from $\mathrm{P}_X^-$. In this section, we introduce domain-agnostic evidence functions applicable to image (Section B.3.1) and tabular datasets (Section B.3.2). While these functions are commonly used as standalone methods for anomaly detection, their role as evidence functions is novel and complementary to our framework. By operating in a transductive setting, they refine the outputs of an AD model initially trained in an inductive setting. Moreover, as shown in Section 5, using these evidence functions solely as anomaly scores does not always yield strong AD performance. However, when integrated into EPHAD, they significantly enhance the performance of a pre-trained model. Finally, the choice of an $T(x)$ is not restricted to AD methods and can be adapted to incorporate domain-specific knowledge for improved effectiveness.

### B.3.1 Evidence for visual datasets

For the evidence function in image-based AD, we propose using Contrastive Language-Image Pre-training (CLIP) (Radford et al., 2021), a robust large-scale framework that learns joint vision-language representations from web-collected image-text pairs. While CLIP has been explored in prior work as a zero-shot AD method (Jeong et al., 2023; Zhou et al., 2024), its performance varies across different datasets. Although CLIP excels in detecting anomalies in real-world image datasets such as CIFAR10, it faces significant challenges when applied to domain-specific datasets, particularly those used for industrial inspection, like MVTec. This limitation stems from the lack of domain-specific knowledge in CLIP's pre-training. In this section, we describe how CLIP is integrated into EPHAD as an evidence function $T(x)$, leveraging its strengths while mitigating its limitations in specialized domains.

Given a dataset $\mathcal{D} := \{(x_j, t_j)\}_{j=1}^n$, CLIP trains an image encoder $e_i$ and a text encoder $e_t$ using contrastive learning (Chen et al., 2020), maximizing the cosine similarity between $e_i(x_j)$ and $e_t(t_j)$ for all $(x_j, t_j) \in \mathcal{D}$. For an input image $x$, CLIP performs zero-shot classification (Radford et al., 2021) by computing a $k$-way categorical distribution over a set of candidate class texts $\mathcal{C} = \{c_1, \ldots, c_k\}$

$$p(c = c_j \mid x; c \in \mathcal{C}) := \frac{\exp\left(\langle e_i(x), e_t(c_j)\rangle / \gamma\right)}{\sum_{s \in \mathcal{C}} \exp\left(\langle e_i(x), e_t(s)\rangle / \gamma\right)},$$

where $\langle \cdot, \cdot \rangle$ denotes the cosine similarity, and $\gamma$ is a temperature parameter that controls the sharpness of the distribution. Pairing class labels $c \in \mathcal{C}$ with prompt templates (e.g., `a photo of a` $[c]$) improves classification accuracy, and aggregating embeddings from multiple prompt variations (e.g., `a cropped photo of a` $[c]$) further enhances performance.

Building on Jeong et al. (2023), we use CLIP as evidence function $T(x)$ in EPHAD. We start by defining two lists of textual prompt templates, $\mathcal{T}_N = \{n_1, \cdots, n_k\}$ and $\mathcal{T}_A = \{a_1, \cdots, a_k\}$, corresponding to normal and anomalous classes, respectively. The list of prompts is provided in Table 3. These templates are dataset-dependent, reflecting subjectivity (e.g., "missing wire" as anomalous for cables). For each label, we generate two lists of prompts for normal and anomalous cases using $\mathcal{T}_N$ and $\mathcal{T}_A$ and compute the mean of text embeddings $t_N$ and $t_A$. Finally, given an input image $x$, the evidence

Table 3: Prompts for CLIP where "c" denotes the category.

| Semantic AD | | Sensory AD | |
|---|---|---|---|
| Normal | Anomalous | Normal | Anomalous |
| `"c"` | `damaged "c"` | `a photo of the number "c"` | `a photo of something` |
| `flawless "c"` | `"c" with flaw` | | |
| `perfect "c"` | `"c" with defect` | | |
| `unblemished "c"` | `"c" with damage` | | |
| `"c" without flaw` | | | |
| `"c" without defect` | | | |
| `"c" without damage` | | | |

$T(x)$ during test-time is computed as:

$$T(x) := \frac{\exp\left(\langle e_i(x), t_N \rangle / \gamma\right)}{\exp\left(\langle e_i(x), t_N \rangle / \gamma\right) + \exp\left(\langle e_i(x), t_A \rangle / \gamma\right)}.$$

One potential concern when using pre-trained models like CLIP is the overlap between their training data and the test samples encountered in downstream tasks. Such overlap could challenge the assumption that test-time statistics are based solely on test data. However, Radford et al. (2021) provides an extensive analysis of this issue and shows that excluding all overlapping samples from CLIP's pre-training corpus leads to only a negligible performance drop. This result suggests that CLIP's effectiveness stems primarily from its generalisation ability rather than memorisation. Accordingly, our experiments emphasise this generalisation property, ensuring that the use of CLIP within our framework remains valid.

### B.3.2 Evidence for tabular datasets

For tabular datasets, we use the output of two classical unsupervised AD methods as evidence functions $T(x)$, namely, Local Outlier Factor (LOF) (Breunig et al., 2000) and Isolation Forest (IForest) (Liu et al., 2012).

**Local Outlier Factor**. To detect anomalies, the local density of a point is compared to that of its $k$-nearest neighbours. Specifically, given a dataset $\mathcal{D} := \{x_j\}_{j=1}^n$, the $k$-distance of a point $x$, denoted as $k$-distance$(x)$, is defined as the distance from $x$ to its $k$-th nearest neighbor.

Based on this, the $k$-distance neighborhood of $x$, denoted as $\mathcal{N}_k(x)$, consists of all points whose distance from $x$ is at most $k$-distance$(x)$. Additionally, the reachability distance of $x$ from a neighbor $x_i$ is computed as reach-dist$_k(x, x_i) = \max\{k\text{-distance}(x), d(x, x_i)\}$, where $d(x, x_i)$ represents the distance between $x$ and $x_i$.

Then, local reachability density (LRD) of $x$ is computed as

$$\text{LRD}_k(x) = \left[ \frac{\sum_{x_i \in N_k(x)} \text{reach-dist}_k(x, x_i)}{|N_k(x)|} \right]^{-1}.$$

Finally, the LOF-based evidence is computed as

$$T(x) = - \frac{\sum_{x_i \in N_k(x)} \frac{\text{LRD}_k(x_i)}{\text{LRD}_k(x)}}{|N_k(x)|}.$$

**Isolation Forest**. Anomalies are identified by recursively partitioning the data using a tree-based method, where features and split values are selected randomly. IForest operates under the assumption that anomalies are more susceptible to isolation due to their sparsity and distinctiveness in the feature space. Given $\mathcal{D}$, IForest constructs multiple isolation trees (ITrees), where each data point $x$ is assigned a depth representing the number of splits required to isolate it, referred to as the *path length*. Specifically, the evidence function $T(x)$ is computed as:

$$T(x) = -2^{-\frac{E(h(x))}{c(n)}},$$

where $h(x)$ is the path length of $x$, i.e., the number of edges traversed from the root node to the leaf node where $x$ is isolated in an ITree. $\mathbb{E}(h(x))$ is the expected path length, i.e., the average path length across multiple ITrees, and $c(n)$ is the average path length of an unsuccessful search.

## B.4 Experimental setup

For training the base AD methods, we use open-source Anomalib and ADBench libraries for experiments with image and tabular datasets, respectively. Our decision to rely on these public libraries was intentional, ensuring transparency and facilitating unbiased comparisons. For the training of each base AD model, we used a single NVIDIA A100 GPU. Then, we run inference using EPHAD on CPU.

# C Extended results

## C.1 Additional experiments on tabular Datasets

Table 4, 5, 6, and 7 summarise the results on a larger set of tabular datasets from ADBench. Each experiment is repeated with three seeds. We can observe that in most cases AD methods benefit from our post-hoc adjustment framework EPHAD, often achieving performance improvements that surpass both the evidence function and the AD method in isolation.

Table 4: Performance of EPHAD on tabular datasets with $10\%$ contamination ratio and LOF as evidence function. Style: AUROC % ($\pm$ SE). Best in **bold**. † represents transductive inference.

| Dataset | LOF† | COPOD Blind | COPOD + EPHAD | DeepSVDD Blind | DeepSVDD + EPHAD | ECOD Blind | ECOD + EPHAD | IForest Blind | IForest + EPHAD | LOF Blind | LOF + EPHAD |
|---|---|---|---|---|---|---|---|---|---|---|---|
| aloi | 72.64 (± 0.1) | 51.46 (± 0.05) | 52.55 (± 0.06) | 54.06 (± 0.54) | 64.36 (± 0.21) | 53.14 (± 0.03) | 54.33 (± 0.05) | 54.05 (± 0.21) | 71.75 (± 2.03) | 73.57 (± 0.1) | 73.62 (± 0.07) |
| annthyroid | 68.53 (± 0.12) | 73.45 (± 0.08) | 73.82 (± 0.08) | 62.69 (± 3.33) | 67.00 (± 2.15) | 76.05 (± 0.11) | 76.31 (± 0.11) | 71.39 (± 0.34) | 70.41 (± 0.13) | 72.12 (± 0.57) | 71.06 (± 0.24) |
| backdoor | 70.43 (± 0.08) | 75.06 (± 0.07) | 78.88 (± 0.08) | 78.34 (± 1.21) | 76.48 (± 0.57) | 83.00 (± 0.09) | 85.48 (± 0.08) | 51.29 (± 1.29) | 70.13 (± 0.12) | 46.65 (± 0.26) | 69.11 (± 0.1) |
| breastw | 46.31 (± 0.92) | 99.46 (± 0.06) | 98.52 (± 0.14) | 98.65 (± 0.05) | 95.13 (± 0.97) | 99.01 (± 0.04) | 97.44 (± 0.03) | 99.46 (± 0.04) | 64.05 (± 1.17) | 73.39 (± 1.35) | 62.4 (± 1.25) |
| celeba | 41.45 (± 0.32) | 72.09 (± 0.01) | 61.86 (± 0.1) | 67.51 (± 3.07) | 55.60 (± 2.13) | 73.99 (± 0.01) | 63.2 (± 0.09) | 40.09 (± 0.83) | 40.32 (± 0.23) | 42.97 (± 0.23) | 40.52 (± 0.38) |
| cover | 52.12 (± 0.1) | 78.70 (± 0.03) | 79.01 (± 0.02) | 75.11 (± 11.37) | 75.74 (± 11.06) | 85.34 (± 0.02) | 85.45 (± 0.02) | 72.59 (± 1.59) | 63.64 (± 0.92) | 22.44 (± 0.1) | 44.20 (± 0.07) |
| fault | 55.00 (± 0.53) | 45.69 (± 0.58) | 45.66 (± 0.57) | 47.34 (± 0.99) | 48.59 (± 0.99) | 47.00 (± 0.4) | 46.87 (± 0.4) | 58.08 (± 0.94) | 55.92 (± 0.68) | 64.41 (± 1.35) | 59.93 (± 0.37) |
| fraud | 45.75 (± 0.13) | 94.39 (± 0.0) | 94.24 (± 0.0) | 89.98 (± 0.97) | 85.1 (± 0.66) | 93.86 (± 0.0) | 93.62 (± 0.01) | 92.95 (± 0.29) | 61.88 (± 0.49) | 33.92 (± 0.34) | 45.26 (± 0.16) |
| glass | 77.52 (± 0.93) | 76.11 (± 0.77) | 79.45 (± 0.95) | 64.52 (± 6.87) | 80.94 (± 3.31) | 67.65 (± 0.44) | 72.59 (± 0.61) | 78.50 (± 1.47) | 79.12 (± 1.01) | 71.79 (± 1.08) | 76.40 (± 0.68) |
| http | 37.65 (± 0.09) | 94.91 (± 0.01) | 90.26 (± 0.04) | 99.17 (± 0.08) | 94.97 (± 0.2) | 92.35 (± 0.02) | 87.88 (± 0.04) | 96.82 (± 0.37) | 69.51 (± 0.62) | 17.85 (± 2.03) | 24.61 (± 0.89) |
| ionosphere | 82.43 (± 0.16) | 79.42 (± 1.03) | 81.67 (± 0.95) | 83.09 (± 0.57) | 84.90 (± 0.17) | 73.04 (± 0.84) | 74.34 (± 0.85) | 89.58 (± 1.57) | 83.50 (± 0.16) | 94.64 (± 0.52) | 89.74 (± 0.55) |
| letter | 83.15 (± 0.73) | 56.71 (± 0.12) | 57.62 (± 0.09) | 50.51 (± 2.54) | 61.26 (± 2.42) | 56.41 (± 0.29) | 57.17 (± 0.29) | 59.84 (± 0.64) | 81.53 (± 0.59) | 85.74 (± 0.54) | 84.84 (± 0.39) |
| lymphography | 99.44 (± 0.26) | 99.52 (± 0.22) | 99.76 (± 0.19) | 98.57 (± 0.74) | 99.53 (± 0.19) | 99.60 (± 0.23) | 99.76 (± 0.19) | 99.76 (± 0.19) | 99.52 (± 0.19) | 98.57 (± 0.59) | 99.36 (± 0.32) |
| mammography | 67.29 (± 0.19) | 89.29 (± 0.05) | 89.28 (± 0.05) | 87.23 (± 0.95) | 87.29 (± 1.22) | 89.38 (± 0.06) | 89.26 (± 0.05) | 80.44 (± 0.29) | 73.93 (± 0.04) | 69.70 (± 0.36) | 72.29 (± 0.18) |
| mnist | 59.63 (± 0.19) | 75.87 (± 0.03) | 75.89 (± 0.03) | 74.26 (± 4.38) | 73.93 (± 4.24) | 72.62 (± 0.05) | 72.64 (± 0.05) | 71.27 (± 0.7) | 62.75 (± 0.16) | 94.55 (± 0.36) | 83.26 (± 0.45) |
| musk | 39.44 (± 0.57) | 91.95 (± 0.32) | 91.91 (± 0.33) | 88.57 (± 5.4) | 87.17 (± 5.87) | 71.84 (± 0.34) | 71.78 (± 0.34) | 89.39 (± 1.88) | 57.06 (± 2.03) | 20.17 (± 0.48) | 32.93 (± 0.04) |
| optdigits | 59.58 (± 0.26) | 62.26 (± 0.24) | 62.49 (± 0.23) | 40.01 (± 10.2) | 46.77 (± 8.53) | 54.04 (± 0.21) | 54.36 (± 0.21) | 40.87 (± 4.5) | 56.80 (± 0.68) | 18.45 (± 0.59) | 50.59 (± 0.07) |
| pendigits | 47.21 (± 0.12) | 88.44 (± 0.2) | 88.38 (± 0.2) | 74.87 (± 9.91) | 72.68 (± 8.72) | 90.63 (± 0.17) | 90.65 (± 0.17) | 81.86 (± 1.48) | 55.56 (± 0.98) | 14.87 (± 0.18) | 37.64 (± 0.13) |
| satellite | 52.90 (± 0.31) | 64.33 (± 0.25) | 64.40 (± 0.25) | 60.59 (± 1.77) | 62.63 (± 1.38) | 57.57 (± 0.16) | 57.61 (± 0.16) | 76.31 (± 0.7) | 63.85 (± 0.4) | 61.01 (± 0.29) | 66.72 (± 0.28) |
| satimage-2 | 52.80 (± 0.15) | 97.03 (± 0.06) | 97.20 (± 0.06) | 92.65 (± 0.46) | 96.16 (± 0.31) | 94.21 (± 0.03) | 94.39 (± 0.02) | 98.91 (± 0.09) | 70.75 (± 0.44) | 24.52 (± 0.87) | 47.14 (± 0.17) |
| shuttle | 55.54 (± 0.11) | 99.26 (± 0.0) | 99.28 (± 0.0) | 97.83 (± 0.91) | 97.78 (± 0.79) | 98.82 (± 0.01) | 98.64 (± 0.01) | 99.57 (± 0.02) | 81.72 (± 0.27) | 99.21 (± 0.01) | 99.69 (± 0.02) |
| smtp | 89.77 (± 0.55) | 79.64 (± 0.01) | 80.56 (± 0.12) | 84.05 (± 0.57) | 86.10 (± 0.5) | 87.98 (± 0.02) | 88.28 (± 0.09) | 89.27 (± 0.88) | 89.80 (± 0.5) | 43.01 (± 1.57) | 89.82 (± 0.27) |
| thyroid | 75.91 (± 0.79) | 88.45 (± 0.35) | 88.71 (± 0.31) | 86.73 (± 3.72) | 88.33 (± 3.15) | 94.91 (± 0.14) | 94.85 (± 0.14) | 93.67 (± 0.27) | 83.42 (± 0.29) | 73.59 (± 1.69) | 77.10 (± 0.53) |
| vowels | 89.10 (± 0.67) | 56.10 (± 0.32) | 58.87 (± 0.34) | 64.47 (± 2.55) | 76.61 (± 1.24) | 54.29 (± 0.06) | 56.82 (± 0.14) | 66.01 (± 0.57) | 88.59 (± 0.65) | 93.04 (± 0.54) | 91.30 (± 0.1) |
| wilt | 64.63 (± 0.72) | 33.45 (± 0.11) | 35.55 (± 0.1) | 35.79 (± 1.97) | 46.44 (± 1.4) | 38.06 (± 0.13) | 39.80 (± 0.15) | 42.92 (± 1.11) | 61.30 (± 0.81) | 81.09 (± 0.41) | 73.37 (± 0.3) |
| wine | 97.57 (± 1.46) | 80.51 (± 1.36) | 86.78 (± 1.96) | 82.26 (± 2.29) | 92.94 (± 1.74) | 67.12 (± 2.04) | 74.97 (± 2.88) | 80.40 (± 3.42) | 97.51 (± 1.51) | 99.94 (± 0.05) | 99.94 (± 0.05) |

Table 5: Performance of EPHAD on tabular datasets with $10\%$ contamination ratio and IForest as evidence function. Style: AUROC % ($\pm$ SE). Best in **bold**. † represents transductive inference.

| Dataset | IForest† | COPOD Blind | COPOD + EPHAD | DeepSVDD Blind | DeepSVDD + EPHAD | ECOD Blind | ECOD + EPHAD | IForest Blind | IForest + EPHAD | LOF Blind | LOF + EPHAD |
|---|---|---|---|---|---|---|---|---|---|---|---|
| aloi | 54.18 (± 0.31) | 51.46 (± 0.05) | 51.48 (± 0.04) | 54.06 (± 0.54) | 54.43 (± 0.51) | 53.14 (± 0.03) | 53.16 (± 0.03) | 54.05 (± 0.21) | 54.26 (± 0.22) | 73.57 (± 0.1) | 69.30 (± 0.18) |
| annthyroid | 78.62 (± 1.01) | 73.45 (± 0.08) | 73.85 (± 0.05) | 62.69 (± 3.33) | 66.63 (± 2.19) | 76.05 (± 0.11) | 76.20 (± 0.09) | 71.39 (± 0.34) | 76.91 (± 0.88) | 72.12 (± 0.57) | 76.67 (± 0.39) |
| backdoor | 67.83 (± 1.69) | 75.06 (± 0.07) | 75.06 (± 0.06) | 78.34 (± 1.21) | 81.43 (± 0.72) | 83.00 (± 0.09) | 82.95 (± 0.09) | 51.29 (± 1.29) | 66.48 (± 1.43) | 46.65 (± 0.26) | 66.23 (± 1.04) |
| breastw | 97.97 (± 0.14) | 99.46 (± 0.06) | 99.46 (± 0.05) | 98.65 (± 0.05) | 98.96 (± 0.04) | 99.01 (± 0.04) | 99.07 (± 0.04) | 99.46 (± 0.04) | 98.98 (± 0.09) | 73.39 (± 1.35) | 81.16 (± 1.08) |
| celeba | 66.62 (± 1.04) | 72.09 (± 0.01) | 72.00 (± 0.01) | 67.51 (± 3.07) | 68.20 (± 2.59) | 73.99 (± 0.01) | 73.87 (± 0.01) | 40.09 (± 0.83) | 60.55 (± 1.07) | 42.97 (± 0.23) | 49.73 (± 0.63) |
| cover | 86.11 (± 1.6) | 78.70 (± 0.03) | 79.01 (± 0.09) | 75.11 (± 11.37) | 77.54 (± 9.82) | 85.34 (± 0.02) | 85.44 (± 0.06) | 72.59 (± 1.59) | 82.94 (± 1.71) | 22.44 (± 0.1) | 76.71 (± 2.42) |
| fault | 52.02 (± 0.18) | 45.69 (± 0.58) | 45.73 (± 0.58) | 47.34 (± 0.99) | 47.89 (± 0.94) | 47.00 (± 0.4) | 44.49 (± 0.39) | 58.08 (± 0.94) | 53.76 (± 0.41) | 64.41 (± 1.35) | 58.97 (± 0.96) |
| fraud | 94.87 (± 0.11) | 94.39 (± 0.0) | 94.40 (± 0.0) | 89.98 (± 0.97) | 92.26 (± 0.5) | 93.86 (± 0.0) | 93.87 (± 0.01) | 92.95 (± 0.29) | 94.60 (± 0.08) | 33.92 (± 0.34) | 85.94 (± 0.34) |
| glass | 77.60 (± 1.77) | 76.11 (± 0.77) | 76.29 (± 0.8) | 64.52 (± 6.87) | 69.28 (± 5.85) | 67.65 (± 0.44) | 68.26 (± 0.52) | 78.50 (± 1.47) | 77.85 (± 1.64) | 71.79 (± 1.08) | 81.23 (± 0.95) |
| http | 99.99 (± 0.0) | 94.91 (± 0.01) | 90.26 (± 0.04) | 99.17 (± 0.08) | 99.24 (± 0.05) | 92.35 (± 0.02) | 94.49 (± 0.07) | 96.82 (± 0.37) | 88.59 (± 0.05) | 17.85 (± 2.03) | 94.04 (± 0.05) |
| ionosphere | 81.80 (± 0.28) | 79.42 (± 1.03) | 79.49 (± 1.0) | 83.09 (± 0.57) | 83.57 (± 0.62) | 73.04 (± 0.84) | 73.21 (± 0.85) | 89.58 (± 1.57) | 85.24 (± 0.63) | 94.64 (± 0.52) | 94.23 (± 0.68) |
| letter | 61.76 (± 0.26) | 56.71 (± 0.12) | 56.76 (± 0.12) | 50.51 (± 2.54) | 52.37 (± 2.32) | 56.41 (± 0.29) | 56.47 (± 0.29) | 59.84 (± 0.64) | 61.35 (± 0.32) | 85.74 (± 0.54) | 80.36 (± 0.32) |
| lymphography | 99.92 (± 0.07) | 99.52 (± 0.22) | 99.52 (± 0.22) | 98.57 (± 0.74) | 99.28 (± 0.41) | 99.60 (± 0.23) | 99.68 (± 0.17) | 99.76 (± 0.19) | 99.84 (± 0.13) | 98.57 (± 0.59) | 99.68 (± 0.26) |
| mammography | 83.98 (± 0.32) | 89.29 (± 0.05) | 89.22 (± 0.04) | 87.23 (± 0.95) | 87.76 (± 0.85) | 89.38 (± 0.06) | 89.24 (± 0.04) | 80.44 (± 0.29) | 83.14 (± 0.17) | 69.70 (± 0.36) | 83.30 (± 0.15) |
| mnist | 75.50 (± 0.08) | 75.87 (± 0.03) | 75.88 (± 0.03) | 74.26 (± 4.38) | 76.20 (± 3.66) | 72.62 (± 0.05) | 72.65 (± 0.05) | 71.27 (± 0.7) | 74.86 (± 0.18) | 94.55 (± 0.36) | 91.46 (± 0.39) |
| musk | 99.29 (± 0.33) | 91.95 (± 0.32) | 92.00 (± 0.32) | 88.57 (± 5.4) | 91.39 (± 4.15) | 71.84 (± 0.34) | 71.92 (± 0.35) | 89.39 (± 1.88) | 98.74 (± 0.21) | 20.17 (± 0.48) | 89.22 (± 2.5) |
| optdigits | 58.65 (± 3.55) | 62.26 (± 0.24) | 62.25 (± 0.26) | 40.01 (± 10.2) | 42.56 (± 9.28) | 54.04 (± 0.21) | 54.09 (± 0.24) | 40.87 (± 4.5) | 53.81 (± 1.83) | 18.45 (± 0.59) | 38.72 (± 2.67) |
| pendigits | 92.04 (± 0.23) | 88.44 (± 0.2) | 88.58 (± 0.21) | 74.87 (± 9.91) | 79.77 (± 8.09) | 90.63 (± 0.17) | 90.73 (± 0.18) | 81.86 (± 1.48) | 90.40 (± 0.11) | 14.87 (± 0.18) | 68.81 (± 1.12) |
| satellite | 64.44 (± 0.57) | 64.33 (± 0.25) | 64.33 (± 0.25) | 60.59 (± 1.77) | 60.84 (± 1.49) | 57.57 (± 0.16) | 57.60 (± 0.16) | 76.31 (± 0.7) | 68.34 (± 0.51) | 61.01 (± 0.29) | 72.19 (± 0.45) |
| satimage-2 | 99.43 (± 0.07) | 97.03 (± 0.06) | 97.06 (± 0.06) | 92.65 (± 0.46) | 95.23 (± 0.06) | 94.21 (± 0.03) | 94.27 (± 0.03) | 98.91 (± 0.09) | 99.41 (± 0.06) | 24.52 (± 0.87) | 92.79 (± 0.16) |
| shuttle | 98.97 (± 0.08) | 99.26 (± 0.0) | 99.28 (± 0.01) | 97.83 (± 0.91) | 98.30 (± 0.78) | 98.82 (± 0.01) | 98.85 (± 0.0) | 99.57 (± 0.02) | 99.46 (± 0.04) | 99.21 (± 0.01) | 99.89 (± 0.01) |
| smtp | 90.95 (± 0.28) | 79.64 (± 0.01) | 81.14 (± 0.06) | 84.05 (± 0.57) | 87.46 (± 0.73) | 87.98 (± 0.02) | 88.41 (± 0.04) | 89.27 (± 0.88) | 90.78 (± 0.3) | 43.01 (± 1.57) | 88.96 (± 0.35) |
| thyroid | 96.65 (± 0.26) | 88.45 (± 0.35) | 89.21 (± 0.32) | 86.73 (± 3.72) | 89.21 (± 2.86) | 94.91 (± 0.14) | 95.06 (± 0.15) | 93.67 (± 0.27) | 96.02 (± 0.18) | 73.59 (± 1.69) | 93.41 (± 0.25) |
| vowels | 72.73 (± 0.8) | 56.10 (± 0.32) | 56.50 (± 0.31) | 64.47 (± 2.55) | 66.27 (± 2.37) | 54.29 (± 0.06) | 54.65 (± 0.06) | 66.01 (± 0.57) | 71.08 (± 0.84) | 93.04 (± 0.54) | 91.68 (± 0.34) |
| wilt | 42.57 (± 1.63) | 33.45 (± 0.11) | 33.70 (± 0.17) | 35.79 (± 1.97) | 36.43 (± 1.88) | 38.06 (± 0.13) | 38.14 (± 0.17) | 42.92 (± 1.11) | 42.66 (± 1.4) | 81.09 (± 0.41) | 71.40 (± 0.64) |
| wine | 58.98 (± 0.68) | 80.51 (± 1.36) | 80.34 (± 1.39) | 82.26 (± 2.29) | 81.07 (± 2.51) | 67.12 (± 2.04) | 67.06 (± 2.08) | 80.40 (± 3.42) | 68.47 (± 2.3) | 99.94 (± 0.05) | 99.72 (± 0.12) |

Table 6: Performance od `EPHAD-Ada` on tabular datasets with $10\%$ contamination ratio and LOF as evidence function. Style: AUROC % ($\pm$ SE). Best in **bold**. † represents transductive inference.

| Dataset | LOF† | COPOD | | DeepSVDD | | ECOD | | IForest | | LOF | |
|---|---|---|---|---|---|---|---|---|---|---|---|
| | | Blind | + EPHAD-Ada | Blind | + EPHAD-Ada | Blind | + EPHAD-Ada | Blind | + EPHAD-Ada | Blind | + EPHAD-Ada |
| aloi | 72.64 (± 0.1) | 51.46 (± 0.05) | **53.65** (± 0.17) | 54.06 (± 0.54) | **70.67** (± 0.22) | 53.14 (± 0.03) | **55.47** (± 0.18) | 54.05 (± 0.31) | **57.49** (± 0.31) | 73.57 (± 0.1) | **73.85** (± 0.05) |
| annthyroid | 68.53 (± 0.12) | 73.45 (± 0.08) | **73.91** (± 0.06) | 62.69 (± 3.33) | **69.27** (± 0.94) | 76.05 (± 0.11) | **76.23** (± 0.06) | 71.39 (± 0.34) | **72.24** (± 0.36) | **72.12** (± 0.57) | 71.79 (± 0.39) |
| backdoor | 70.43 (± 0.08) | **75.06** (± 0.07) | 75.04 (± 0.07) | **78.34** (± 1.21) | 78.31 (± 1.21) | **83.0** (± 0.09) | 82.99 (± 0.09) | **51.29** (± 1.29) | **51.29** (± 1.29) | 46.65 (± 0.26) | **61.97** (± 1.74) |
| breastw | 46.31 (± 0.92) | **99.46** (± 0.06) | 97.73 (± 0.06) | **98.65** (± 0.05) | 92.94 (± 1.83) | **99.01** (± 0.04) | 96.87 (± 0.26) | **99.46** (± 0.04) | 97.71 (± 0.21) | **73.39** (± 1.35) | 66.12 (± 1.49) |
| celeba | 41.45 (± 0.32) | **72.09** (± 0.01) | 70.85 (± 0.07) | **67.51** (± 3.07) | 64.46 (± 3.06) | **73.99** (± 0.01) | 72.89 (± 0.06) | **40.09** (± 0.83) | 37.54 (± 0.87) | **42.97** (± 0.23) | 40.96 (± 0.34) |
| cover | 52.12 (± 0.1) | 78.7 (± 0.03) | **79.57** (± 0.01) | 75.11 (± 11.37) | **75.58** (± 10.82) | 85.34 (± 0.02) | **85.45** (± 0.01) | 72.59 (± 1.59) | **73.15** (± 1.57) | 22.44 (± 0.1) | **36.78** (± 0.23) |
| fault | 55.0 (± 0.53) | 45.69 (± 0.58) | **45.79** (± 0.58) | 47.34 (± 0.99) | **50.25** (± 0.65) | **47.0** (± 0.4) | 46.81 (± 0.39) | **58.08** (± 0.94) | 57.61 (± 0.9) | **64.41** (± 1.35) | 61.29 (± 0.9) |
| fraud | 45.75 (± 0.13) | **94.39** (± 0.0) | 94.38 (± 0.01) | **89.98** (± 0.97) | 89.93 (± 0.97) | **93.86** (± 0.0) | 93.84 (± 0.01) | **92.95** (± 0.29) | 92.94 (± 0.29) | 33.92 (± 0.34) | **43.01** (± 0.84) |
| glass | 77.52 (± 0.93) | 76.11 (± 0.77) | **81.77** (± 1.28) | 64.52 (± 6.87) | **80.94** (± 2.52) | 67.65 (± 0.44) | **78.43** (± 1.72) | 78.5 (± 1.47) | **83.15** (± 1.86) | 71.79 (± 1.08) | **75.67** (± 0.75) |
| http | 37.65 (± 0.09) | **94.91** (± 0.01) | **94.91** (± 0.01) | **99.17** (± 0.08) | **99.17** (± 0.08) | **92.35** (± 0.02) | **92.35** (± 0.02) | **96.82** (± 0.37) | **96.82** (± 0.37) | 17.85 (± 2.03) | **18.31** (± 1.96) |
| ionosphere | 82.43 (± 0.16) | 79.42 (± 1.03) | **84.15** (± 0.38) | 83.09 (± 0.57) | **85.03** (± 0.25) | 73.04 (± 0.84) | **78.14** (± 0.49) | 89.58 (± 1.57) | **90.05** (± 1.22) | **94.64** (± 0.52) | 91.85 (± 0.68) |
| letter | 83.15 (± 0.73) | 56.71 (± 0.12) | **71.03** (± 0.99) | 50.51 (± 2.54) | **65.9** (± 2.88) | 56.41 (± 0.29) | **70.15** (± 1.15) | 59.84 (± 0.64) | **71.38** (± 0.86) | **85.74** (± 0.54) | 85.31 (± 0.36) |
| lymphography | 99.44 (± 0.26) | 99.52 (± 0.22) | **99.84** (± 0.13) | 98.57 (± 0.74) | **99.45** (± 0.24) | 99.6 (± 0.23) | **99.84** (± 0.13) | **99.76** (± 0.19) | 99.71 (± 0.21) | 98.57 (± 0.59) | **99.28** (± 0.39) |
| mammography | 67.29 (± 0.19) | **89.29** (± 0.05) | 89.23 (± 0.05) | **87.23** (± 0.95) | 87.11 (± 0.95) | **89.38** (± 0.06) | 89.32 (± 0.06) | **80.44** (± 0.29) | 80.37 (± 0.29) | 69.7 (± 0.36) | **73.82** (± 0.06) |
| mnist | 59.63 (± 0.19) | **75.87** (± 0.03) | 74.27 (± 0.12) | **74.26** (± 4.38) | 61.3 (± 0.69) | **72.62** (± 0.05) | 70.83 (± 0.19) | **71.27** (± 0.7) | 70.59 (± 0.56) | **94.55** (± 0.36) | 88.51 (± 0.49) |
| musk | 39.44 (± 0.57) | **91.95** (± 0.32) | 85.69 (± 0.87) | **88.57** (± 5.4) | 81.67 (± 7.04) | **71.84** (± 0.34) | 65.84 (± 0.57) | **89.39** (± 1.88) | 82.04 (± 3.01) | 20.17 (± 0.48) | **28.5** (± 0.74) |
| optdigits | 59.58 (± 0.26) | 62.26 (± 0.24) | **65.13** (± 0.22) | 40.01 (± 10.2) | **58.64** (± 0.91) | 54.04 (± 0.21) | **58.99** (± 0.28) | 40.87 (± 4.5) | **48.26** (± 3.08) | 18.45 (± 0.59) | **42.52** (± 0.89) |
| pendigits | 47.21 (± 0.12) | **88.44** (± 0.2) | 87.09 (± 0.22) | **74.87** (± 9.91) | 74.08 (± 9.18) | 90.63 (± 0.17) | **92.35** (± 0.02) | **81.86** (± 1.48) | 79.5 (± 1.5) | 14.87 (± 0.18) | **30.16** (± 1.01) |
| satellite | 52.9 (± 0.31) | 64.33 (± 0.25) | **66.71** (± 0.3) | 60.59 (± 1.77) | **63.44** (± 1.53) | 57.57 (± 0.16) | **59.69** (± 0.2) | **76.31** (± 0.7) | 76.08 (± 0.46) | 61.01 (± 0.29) | **66.71** (± 0.29) |
| satimage-2 | 52.8 (± 0.15) | 97.03 (± 0.06) | **98.53** (± 0.07) | 92.65 (± 0.46) | **96.14** (± 0.32) | 94.21 (± 0.03) | **96.41** (± 0.07) | **98.91** (± 0.09) | 98.16 (± 0.3) | 24.52 (± 0.87) | **41.8** (± 0.72) |
| shuttle | 55.54 (± 0.11) | **99.26** (± 0.0) | **99.26** (± 0.01) | **97.83** (± 0.91) | 89.97 (± 1.95) | **99.82** (± 0.01) | 99.8 (± 0.01) | **99.57** (± 0.02) | **99.57** (± 0.02) | 99.21 (± 0.01) | **99.82** (± 0.0) |
| smtp | 89.77 (± 0.55) | 79.64 (± 0.01) | **79.69** (± 0.01) | **84.05** (± 0.57) | 83.73 (± 0.4) | 87.98 (± 0.02) | **88.0** (± 0.03) | **89.27** (± 0.88) | **89.27** (± 0.88) | 43.01 (± 1.57) | **63.18** (± 2.15) |
| thyroid | 75.91 (± 0.79) | 88.45 (± 0.35) | **88.54** (± 0.25) | **86.73** (± 3.72) | 85.53 (± 3.6) | **94.91** (± 0.14) | 94.06 (± 0.1) | **93.67** (± 0.27) | 93.11 (± 0.19) | 73.59 (± 1.69) | **76.74** (± 0.69) |
| vowels | 89.1 (± 0.67) | 56.1 (± 0.32) | **65.07** (± 0.52) | 64.47 (± 2.55) | **82.12** (± 0.9) | 54.29 (± 0.06) | **75.39** (± 0.91) | 60.01 (± 0.57) | **80.76** (± 0.6) | **93.04** (± 0.54) | 91.85 (± 0.12) |
| wilt | 64.63 (± 0.72) | 33.45 (± 0.11) | **38.4** (± 0.73) | 35.79 (± 1.97) | **59.53** (± 1.5) | 38.06 (± 0.13) | **42.06** (± 0.48) | 42.92 (± 1.11) | **47.27** (± 0.39) | **81.09** (± 0.41) | 76.62 (± 0.9) |
| wine | 97.57 (± 1.46) | 80.51 (± 1.36) | **93.96** (± 1.66) | 82.26 (± 2.29) | **93.96** (± 1.77) | 67.12 (± 2.04) | **89.27** (± 2.95) | 80.4 (± 3.42) | **93.56** (± 2.15) | **99.94** (± 0.05) | **99.94** (± 0.05) |

Table 7: Performance od `EPHAD-Ada` on tabular datasets with $10\%$ contamination ratio and IForest as evidence function. Style: AUROC % ($\pm$ SE). Best in **bold**. † represents transductive inference.

| Dataset | IForest† | COPOD | | DeepSVDD | | ECOD | | IForest | | LOF | |
|---|---|---|---|---|---|---|---|---|---|---|---|
| | | Blind | + EPHAD-Ada | Blind | + EPHAD-Ada | Blind | + EPHAD-Ada | Blind | + EPHAD-Ada | Blind | + EPHAD-Ada |
| aloi | 54.18 (± 0.31) | 51.46 (± 0.05) | **52.42** (± 0.1) | 54.06 (± 0.54) | **54.21** (± 0.32) | 53.14 (± 0.03) | **53.72** (± 0.11) | 54.05 (± 0.21) | **54.27** (± 0.12) | **73.57** (± 0.1) | 62.1 (± 0.6) |
| annthyroid | 78.62 (± 1.01) | 73.45 (± 0.08) | **77.13** (± 0.43) | 62.69 (± 3.33) | **77.91** (± 0.79) | 76.05 (± 0.11) | **77.84** (± 0.5) | 71.39 (± 0.34) | **75.66** (± 0.69) | 72.12 (± 0.57) | **77.98** (± 0.7) |
| backdoor | 67.83 (± 1.69) | **75.06** (± 0.07) | 73.35 (± 0.54) | **78.34** (± 1.21) | 72.05 (± 1.0) | **83.0** (± 0.09) | 78.27 (± 0.46) | 51.29 (± 1.29) | **61.25** (± 0.57) | 46.65 (± 0.26) | **67.81** (± 1.68) |
| breastw | 97.97 (± 0.14) | **99.46** (± 0.06) | 99.29 (± 0.03) | 98.65 (± 0.05) | **98.85** (± 0.06) | 99.01 (± 0.04) | **99.1** (± 0.06) | **99.46** (± 0.04) | 97.71 (± 0.21) | 73.39 (± 1.35) | **92.91** (± 0.52) |
| celeba | 66.62 (± 1.04) | **72.09** (± 0.01) | 69.51 (± 0.51) | 67.51 (± 3.07) | **68.83** (± 1.07) | **73.99** (± 0.01) | 70.52 (± 0.49) | 40.09 (± 0.83) | **54.01** (± 0.64) | 42.97 (± 0.23) | **60.04** (± 0.95) |
| cover | 86.11 (± 1.6) | 78.7 (± 0.03) | | 75.11 (± 11.37) | **84.6** (± 3.81) | 85.34 (± 0.02) | **86.56** (± 0.94) | 72.59 (± 1.59) | **80.23** (± 1.78) | 22.44 (± 0.1) | **80.33** (± 2.26) |
| fault | 52.02 (± 0.18) | 45.69 (± 0.58) | **48.69** (± 0.37) | 47.34 (± 0.99) | **51.44** (± 0.25) | 47.0 (± 0.4) | **49.3** (± 0.31) | **58.08** (± 0.94) | 55.16 (± 0.61) | **64.41** (± 1.35) | 52.81 (± 0.11) |
| fraud | 94.87 (± 0.11) | 94.39 (± 0.0) | **94.81** (± 0.07) | 89.98 (± 0.97) | **94.84** (± 0.11) | 93.86 (± 0.0) | **94.63** (± 0.01) | 92.95 (± 0.29) | **94.32** (± 0.09) | 33.92 (± 0.34) | **94.86** (± 0.1) |
| glass | 77.6 (± 1.77) | 76.11 (± 0.77) | **77.78** (± 1.44) | 64.52 (± 6.87) | **76.33** (± 2.04) | 67.65 (± 0.44) | **74.08** (± 1.16) | **78.5** (± 1.47) | 77.96 (± 1.6) | 71.79 (± 1.08) | **83.73** (± 1.51) |
| http | 99.99 (± 0.0) | 94.91 (± 0.01) | **99.45** (± 0.03) | 99.17 (± 0.08) | **99.52** (± 0.01) | 92.35 (± 0.02) | **99.25** (± 0.01) | 96.82 (± 0.37) | **99.37** (± 0.01) | 17.85 (± 2.03) | **99.98** (± 0.0) |
| ionosphere | 81.8 (± 0.28) | 79.42 (± 1.03) | **81.84** (± 0.61) | 83.09 (± 0.57) | **83.72** (± 0.5) | 73.04 (± 0.84) | **78.3** (± 0.51) | **89.58** (± 1.57) | 86.62 (± 0.87) | **94.64** (± 0.52) | 89.88 (± 0.6) |
| letter | 61.76 (± 0.26) | 56.71 (± 0.12) | **59.61** (± 0.28) | 50.51 (± 2.54) | **56.79** (± 1.64) | 56.41 (± 0.29) | **59.37** (± 0.28) | 59.84 (± 0.64) | **60.93** (± 0.4) | **85.74** (± 0.54) | 79.09 (± 0.62) |
| lymphography | 99.92 (± 0.07) | 99.52 (± 0.22) | **99.76** (± 0.19) | 98.57 (± 0.74) | **99.92** (± 0.07) | 99.6 (± 0.23) | **99.76** (± 0.19) | **99.76** (± 0.19) | **99.76** (± 0.19) | 98.57 (± 0.59) | **99.84** (± 0.13) |
| mammography | 83.98 (± 0.32) | **89.29** (± 0.05) | 87.06 (± 0.13) | **87.23** (± 0.95) | 84.27 (± 0.27) | **89.38** (± 0.06) | 87.51 (± 0.12) | 80.44 (± 0.29) | **82.57** (± 0.09) | 69.7 (± 0.36) | **83.91** (± 0.31) |
| mnist | 75.5 (± 0.08) | 75.87 (± 0.03) | **76.47** (± 0.02) | 74.26 (± 4.38) | **76.04** (± 0.24) | 72.62 (± 0.05) | **74.8** (± 0.04) | 71.27 (± 0.7) | **73.83** (± 0.38) | **94.55** (± 0.36) | 90.56 (± 0.41) |
| musk | 99.29 (± 0.33) | 91.95 (± 0.32) | **97.37** (± 0.45) | 88.57 (± 5.4) | **97.34** (± 1.28) | 71.84 (± 0.34) | **90.7** (± 1.37) | 89.39 (± 1.88) | **96.77** (± 0.15) | 20.17 (± 0.48) | **77.73** (± 2.96) |
| optdigits | 58.65 (± 3.55) | **62.26** (± 0.24) | 60.85 (± 1.93) | 40.01 (± 10.2) | **58.15** (± 3.71) | 54.04 (± 0.21) | **57.14** (± 2.15) | 40.87 (± 4.5) | **50.42** (± 1.43) | 18.45 (± 0.59) | **38.14** (± 3.6) |
| pendigits | 92.04 (± 0.23) | 88.44 (± 0.2) | **90.69** (± 0.26) | 74.87 (± 9.91) | **90.09** (± 2.25) | 90.63 (± 0.17) | **92.11** (± 0.18) | 81.86 (± 1.48) | **88.36** (± 0.42) | 14.87 (± 0.18) | **79.82** (± 0.64) |
| satellite | 64.44 (± 0.57) | 64.33 (± 0.25) | **64.72** (± 0.3) | 60.59 (± 1.77) | **62.43** (± 0.92) | 57.57 (± 0.16) | **61.03** (± 0.21) | **76.31** (± 0.7) | 69.99 (± 0.41) | 61.01 (± 0.29) | **72.04** (± 0.39) |
| satimage-2 | 99.43 (± 0.07) | 97.03 (± 0.06) | **98.75** (± 0.06) | 92.65 (± 0.46) | **98.19** (± 0.24) | 94.21 (± 0.03) | **97.87** (± 0.08) | 98.91 (± 0.09) | **99.31** (± 0.07) | 24.52 (± 0.87) | **95.39** (± 0.14) |
| shuttle | 98.97 (± 0.08) | 99.26 (± 0.0) | **99.42** (± 0.05) | 97.83 (± 0.91) | **99.08** (± 0.06) | **99.82** (± 0.01) | 99.08 (± 0.06) | 99.57 (± 0.02) | **99.79** (± 0.03) | 99.21 (± 0.01) | **99.79** (± 0.03) |
| smtp | 90.95 (± 0.28) | 79.64 (± 0.01) | **88.06** (± 0.17) | 84.05 (± 0.57) | **91.05** (± 0.32) | 87.98 (± 0.02) | **90.21** (± 0.17) | 89.27 (± 0.88) | **90.49** (± 0.4) | 43.01 (± 1.57) | **90.9** (± 0.22) |
| thyroid | 96.65 (± 0.26) | 88.45 (± 0.35) | **94.35** (± 0.26) | 86.73 (± 3.72) | **95.8** (± 0.48) | 94.91 (± 0.14) | **96.2** (± 0.21) | 93.67 (± 0.27) | **95.5** (± 0.14) | 73.59 (± 1.69) | **95.67** (± 0.1) |
| vowels | 72.73 (± 0.8) | 56.1 (± 0.32) | **65.07** (± 0.52) | 64.47 (± 2.55) | **70.74** (± 1.46) | 54.29 (± 0.06) | **64.37** (± 0.67) | 66.01 (± 0.57) | **69.74** (± 0.2) | **93.04** (± 0.54) | 90.01 (± 0.24) |
| wilt | 42.57 (± 1.63) | 33.45 (± 0.11) | **37.63** (± 0.95) | 35.79 (± 1.97) | **41.82** (± 1.71) | 38.06 (± 0.13) | **39.62** (± 0.84) | **42.92** (± 1.11) | 42.76 (± 1.28) | **81.09** (± 0.41) | 61.95 (± 2.3) |
| wine | 58.98 (± 0.68) | **80.51** (± 1.36) | 74.58 (± 1.48) | **82.26** (± 2.29) | 73.34 (± 2.89) | **67.12** (± 2.04) | 63.73 (± 0.96) | **80.4** (± 3.42) | 73.62 (± 3.11) | **99.94** (± 0.05) | 97.29 (± 0.64) |

## C.2 Experiments on industrial use case

**CSP plant dataset**. For the industrial setting, we utilise the simulated dataset introduced by Patra et al. (2024), which is generated by training a variational autoencoder on real-world data collected from an operational CSP plant. The dataset consists of thermal images of solar panels captured using infrared (IR) cameras, distinguishing it from the semantic and sensory anomaly datasets, as the images lack semantic structure and do not depict specific objects.

**Baseline AD method**. We evaluate the performance of the forecasting-based anomaly detection method `ForecastAD`, as proposed by the original authors, both with and without the integration of `EPHAD`. All experiments are conducted using the original implementation provided by the authors.

**Rule-based evidence**. Foundation models, such as CLIP, which were previously used in our experiments on image datasets, are not applicable in specialised applications, such as detecting anomalous behaviour in solar power plants, due to the lack of semantic content in thermal images. This makes zero-shot methods like WinCLIP and AnoCLIP inapplicable. In contrast, while EPHAD can in-

corporate evidence from foundation models like CLIP, it also allows the seamless integration of domain-specific knowledge. To compute evidence, we utilise two of the four rules proposed by Patra et al. (2024) that indicate normal operational behaviour of the CSP plant. The first rule (**R1**) is based on the *difference between consecutive images*. Under normal conditions, the plant's temperature is expected to remain relatively stable; therefore, substantial deviations from one image to the next suggest potential anomalies. To quantify this, pixel-wise squared differences are computed between every pair of consecutive images, and the 95th percentile of these differences is extracted as the representative evidence for each pair. The second rule (**R2**) involves the *difference from the average daily temperature*. Here, samples with average temperatures significantly diverging from the typical daily average could indicate anomalous behaviour. For this, the mean temperature of each day is first determined, and then the absolute difference between each image's average temperature and that day's mean is computed to serve as the evidence.

Table 8: Performance on CSP plant dataset.

| Setting | Method | AUROC (± SE) |
|---|---|---|
| Clean | `ForecastAD` | 94.91 (±0.09) |
| Evidence | Rule-based (R1, R2) | 69.46 (± 0.0) |
| Contaminated | `ForecastAD` | 90.45 (± 0.8) |
| ($\epsilon = 0.1$) | + `EPHAD` | 93.51 (± 0.45) |
| | + `EPHAD-Ada` | **93.57** (± 0.43) |

**Results**. The results presented in Table 8 underscore the effectiveness and adaptability of our approach. Under a 10% contamination setting, the baseline method `ForecastAD` experiences a performance drop of approximately 5%. However, by incorporating domain-specific rules R1 and R2 as sources of evidence using `EPHAD` and further using `EPHAD-Ada`, the performance nearly matches that on the clean dataset. It emphasises the value of leveraging structured, context-aware evidence to enhance the detection of anomalies. Importantly, foundation models like CLIP are unsuitable in this context due to the lack of semantic content in thermal imagery, rendering zero-shot approaches such as WinCLIP (Jeong et al., 2023) and AnoCLIP (Zhou et al., 2024) ineffective. `EPHAD` addresses this limitation by providing a flexible framework that integrates both powerful foundation models, where applicable, and domain-specific knowledge when necessary. This versatility enables `EPHAD` to deliver robust performance across diverse real-world anomaly detection tasks while maintaining efficiency and ease of deployment.

### C.3 Comparison against LOE and SoftPatch

To ensure a comprehensive evaluation, we compare the performance of our proposed post-hoc framework against SoftPatch (Jiang et al., 2022) and both variants of LOE (Qiu et al., 2022). However, it is important to note that, unlike our approach, both SoftPatch and LOE modify the training process to account for contamination, making it inapplicable to pre-trained networks without access to the training dataset and pipeline, which is our main focus.

Table 9: Comparison with LOE (AUROC %)

| | Method | Semantic AD | | | | Sensory AD | | |
|---|---|---|---|---|---|---|---|---|
| | | MNIST | FMNIST | CIFAR10 | SVHN | MVTec | MPDD | ViSA |
| | CLIP | 71.15 | 95.63 | 98.63 | 58.46 | 86.34 | 60.02 | 74.47 |
| NTL | Blind | 90.15 | 89.01 | 90.79 | **61.82** | 78.13 | 80.41 | 61.95 |
| NTL | Refine | 91.35 | 91.37 | 92.79 | 61.78 | 82.54 | 87.32 | 65.63 |
| NTL | LOE-Hard | 86.89 | 90.53 | 93.10 | 53.86 | 79.28 | 83.34 | **78.82** |
| NTL | LOE-Soft | **91.56** | 92.89 | 94.71 | 61.69 | 85.46 | **92.31** | 74.5 |
| | `EPHAD` | 78.96 | **95.99** | **98.65** | 57.64 | **86.20** | 59.88 | 74.22 |

First, for comparison with LOE, we conduct experiments using the Neural Transformation Learning-based (NTL) AD method (Qiu et al., 2021) and evaluate it under four configurations: "Blind", "Refine", LOE-Hard and LOE-Soft. Additionally, we follow the same setup as LOE by extracting image features using pre-trained ResNet152 and WideResNet50 for semantic and sensory datasets, respectively, which are then used to train NTL. The results, summarised in Table 9, show that given a good evidence function, i.e. the performance of the evidence is better than the "Blind" configuration, our simple test-time framework outperforms LOE. Results on MVTec, CIFAR10, FMIST, and SVHN are examples of this behaviour. Also, on the ViSA dataset, the performance improves over the "Blind" and "Refine" configurations. In the converse situations where the performance of the evidence is lower than the "Blind" configuration, we observe a reduction in performance which can be accounted for by putting more emphasis on the AD model by adjusting $\beta$.

Now, we compare it against SoftPatch, an approach built upon PatchCore ([Roth et al., 2022](#)). SoftPatch enhances PatchCore by incorporating traditional anomaly detection (AD) techniques to refine the memory bank, specifically by identifying and re-weighting patches based on their outlier scores during training. While this strategy improves performance, it introduces a strong dependency on the choice of AD method and increases the computational burden of the training pipeline.

For a fair comparison, we adopt the Local Outlier Factor (LOF) as the AD method, as it has been empirically found to be the most effective for SoftPatch. As shown in Table 10, our method, EPHAD, achieves competitive results despite being a fully post-hoc approach that requires no modification to the training process. Crucially, while SoftPatch is tailored for memory-bank-based methods, EPHAD is inherently model-agnostic and can be seamlessly applied to any combination of a pre-trained model and an evidence function. This versatility highlights EPHAD's broad applicability and practical utility across a diverse range of settings.

Table 10: Comparison with SoftPatch

| | Method | Sensory AD | | |
| | | MVTec | MPDD | ViSA |
|---|---|---|---|---|
| | CLIP | 86.34 | 60.02 | 74.47 |
| PatchCore | Blind | 70.02 | 51.41 | 19.91 |
| | SoftPatch | **90.40** | **67.00** | **86.54** |
| | EPHAD | 86.45 | 60.58 | 62.94 |

## C.4 Ablation on $\epsilon$ and $\beta$

Extended ablation on $\epsilon$ and $\beta$ can be found in Figure 3, 4. We can make similar conclusions as discussed above in Section 5.3.

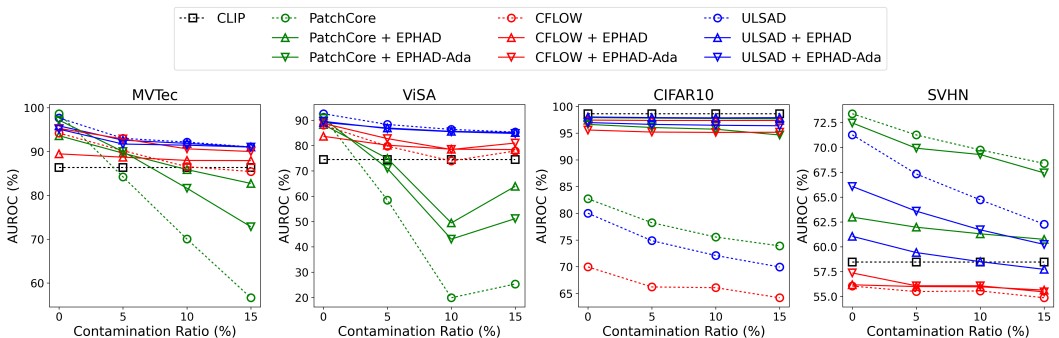

Figure 3: Ablation on $\epsilon$.

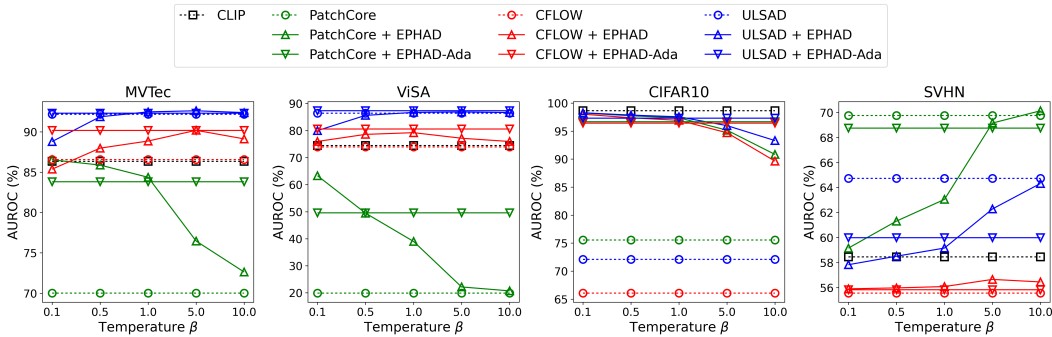

Figure 4: Ablation on $\beta$.

## C.5 Effect of test set size $n$

The performance of our proposed framework, EPHAD, is influenced by both the pre-trained AD method and the evidence function. While the pre-trained AD method is affected only by the training data, for the evidence function, we evaluated two scenarios: (1) When using foundation models such as CLIP, the evidence function remains independent of the test sample distribution. (2) When employing traditional AD methods like Isolation Forest or Local Outlier Factor, the evidence function

relies on the local density of test samples, meaning that an insufficient number of test samples could lead to less informative evidence which can be accounted for in EPHAD by adjusting the temperature parameter $\beta$. In Figure 5, we analyse the impact of varying the proportion of anomalies in the test set, which exhibits consistent improvements across all tested settings.

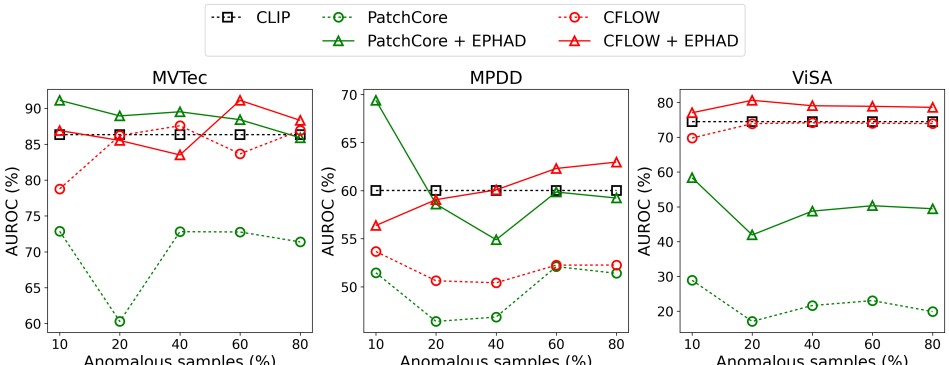

Figure 5: Ablation on varying proportion of anomalies in the test set.

