# OpenReview forum: "An Evidence-Based Post-Hoc Adjustment Framework for Anomaly Detection Under Data Contamination"
_NeurIPS.cc/2025/Conference — NeurIPS 2025 spotlight_

### Official Review · Reviewer_qVBz · 2025-06-16

**Clarity:** 3
**Significance:** 3
**Originality:** 3
**Rating:** 5
**Confidence:** 2

**Summary:**

This paper proposes a test-time framework to adapt the scores learnt by anomaly detection models trained on contaminated training data using Bayesian inference. The method is tested on a variety of AD datasets and shows consistent improvement over anomaly detection performance with the base anomaly scores.

**Questions:**

How does this approach differ from and compare with other methods which address data contamination in anomaly detection?

**Ethical Concerns:**

["NO or VERY MINOR ethics concerns only"]

**Final Justification:**

I am satisfied with the authors original paper as well as subsequent rebuttal.

**Limitations:**

Limitations are addressed but are fairly surface-level.

**Quality:**

4

**Strengths And Weaknesses:**

The motivation for the methodology is well explained and clear.

The experiments are comprehensive across a wide variety of datasets, of different modalities and contexts (synthetic, tabular, image, industrial) and across a variety of base AD methods.

The methodology shows consistent and noticeable improvement in AD performance.

No significant weaknesses to mention, except the authors could make more of an effort to provide context for this methodology within AD research. What other similar methods have been taken along this approach? How does this methodology improve upon those?

---

> ### Author Rebuttal · Authors · 2025-07-31
>
> We thank the reviewer for their constructive feedback. We are happy to note that the reviewer finds our proposed method well motivated, clearly explained and empirically effective in improving AD performance. We address the reviewer's concerns in the points below.
>
> **[W1, Q1] How does this approach differ from and compare with other methods which address data contamination in anomaly detection?** While we acknowledge prior work on anomaly detection with contaminated data, to the best of our knowledge, no existing framework offers post-hoc adaptation in the same manner as EPHAD. Unlike previous methods, EPHAD does not require access to the pre-trained anomaly detection (AD) model’s weights or training pipeline, making it fundamentally different in its approach. Our primary focus in the experiments is to demonstrate the effectiveness, versatility, and applicability of EPHAD, a simple yet powerful Bayesian inference-based framework, by evaluating its relative performance against the pre-trained model and the evidence function in isolation.
>
> Nonetheless, for completeness, we have compared EPHAD to two existing frameworks: ``Refine'' [1] and Latent Outlier Exposure (LOE) [2], as reported in Table 2 of our paper. While Refine iteratively identifies and removes anomalies during training, LOE incorporates identified anomalies using a contrastive loss.
>
> [1] Yoon et al., "Self-supervise, Refine, Repeat: Improving Unsupervised Anomaly Detection" TMLR, 2022.
> [2] Qiu et al., "Latent outlier exposure for anomaly detection with contaminated data." ICML, 2022.

---

> ### Author Response · Authors · 2025-08-05
>
> Dear Reviewer qVBz,
>
> Thank you once again for your time and insightful feedback, which has helped us improve our paper. We hope our rebuttal has addressed your concerns regarding the comparison of EPHAD with existing methods that address contamination in anomaly detection.
>
> If you have any further questions or comments, please let us know before the discussion period ends. We are happy to clarify any remaining points.
>
> Best regards,
> The Authors

---

> > ### Comment · Reviewer_qVBz · 2025-08-07
> >
> > I am satisfied with the authors' responses.

---

> > > ### Author Response · Authors · 2025-08-07
> > >
> > > Thank you for your time. We are happy to note that our response addressed your concerns.

---

### Official Review · Reviewer_qm8P · 2025-06-29

**Clarity:** 3
**Significance:** 3
**Originality:** 3
**Rating:** 5
**Confidence:** 4

**Summary:**

This paper focuses on anomaly detection under data contamination. Through an evidence function, this paper proposes EPHAD, an inference-time adaptation framework that updates the outputs of AD models trained on contaminated datasets using evidence gathered at inference. Experiments are conducted on several popular datasets and show good performance.

**Questions:**

See the weakness. The concerns are mainly from the design of the evidence function.

**Ethical Concerns:**

["NO or VERY MINOR ethics concerns only"]

**Final Justification:**

Thank you for the response. As the authors have addressed all my concerns, I will raise my score to accept.

**Limitations:**

See above.

**Quality:**

3

**Strengths And Weaknesses:**

1. The design of the Evidence Function constitutes the most critical part of the paper and decisively influences the results. However, the main text downplays this aspect. After reading the entire paper, it remains unclear how the Evidence Function was designed. This design should be discussed in detail as a central component of the main text.
2. The supplementary materials describe the Evidence Function, revealing a severe contradiction in the paper's methodology. CLIP inherently supports zero-shot anomaly detection (e.g., WinCLIP [a], MVFA [b]), which is unaffected by contamination in training data. This suggests the authors merely combined methods that inherently ignore anomalous noise. Moreover, no experimental comparison was made with these CLIP-based approaches.
3. Does using CLIP as the Evidence Function introduce data leakage? For instance, WinCLIP excels on MVTec but underperforms on datasets like MPDD. This is reflected in Table 1: when EPHAD is combined with RD and ULSAD, performance on MPDD deteriorates compared to not using EPHAD at all. Also, how EPHAD performs when it is used on datasets in other domains, such as medical data used in MVFA.
4. The Introduction claims that existing methods rely on 'the contamination ratio.' However, these papers never mention such a ratio. Some use hyperparameters to set the proportion of anomalous noise removal, but this proportion ≠ the contamination ratio. Experiments confirm these hyperparameters minimally affect results, contradicting the authors' overstated dependency claims.

[a] WinCLIP: Zero-/Few-Shot Anomaly Classification and Segmentation. CVPR 2023.
[b] Adapting Visual-Language Models for Generalizable Anomaly Detection in Medical Images. CVPR 2024.

---

> ### Author Rebuttal · Authors · 2025-07-31
>
> We thank the reviewer for their constructive feedback. We address the reviewer's concerns in the points below.
>
> **[W1] It remains unclear how the Evidence Function was designed.** In our current draft, we provide additional details on the evidence functions in Appendix B.3. Notably, while EPHAD can incorporate evidence from foundation models like CLIP, it also allows the seamless integration of domain-specific knowledge. To demonstrate this versatility, we have shown CLIP as evidence for image-based experiments and LOF as evidence for tabular datasets. Furthermore, in the appendix, we have shown domain-specific rules as evidence. For this, we examine the AD method, forecastAD, for thermal images introduced by [1] and compare the performance of forecastAD with and without EPHAD under $10\%$ contamination. The results presented in Table 5 of our current draft show that while forecastAD experiences a performance drop of approximately $5\%$, incorporating domain-specific rules R1 and R2, as outlined in [1], provides evidence for EPHAD, which recovers about $2.5\%$ of the lost performance. Moreover, we show this generality beyond image data by applying EPHAD to tabular datasets using traditional AD methods such as LOF and Isolation Forest as the evidence function.
>
> **[W2, W3] CLIP inherently supports zero-shot anomaly detection (e.g., WinCLIP [a], MVFA [b]), which is unaffected by contamination in training data. Does using CLIP as the Evidence Function introduce data leakage?** While CLIP has been employed in prior work for zero-shot anomaly detection, our approach does not rely on CLIP as a standalone anomaly detector. Instead, we leverage its image-text matching capabilities to provide auxiliary evidence that is integrated within a broader and more flexible framework. It is also worth highlighting that foundation models like CLIP are not applicable in specialised applications such as the detection of anomalous behaviour in solar power plants due to the lack of semantic content in thermal images. This makes zero-shot methods like WinCLIP and AnoCLIP inapplicable. In contrast, while EPHAD can incorporate evidence from foundation models like CLIP, it also allows the seamless integration of domain-specific knowledge as shown in Appendix C.1 of our current draft. This makes our framework more adaptable and broadly applicable across domains.
>
> For the experiments on image-based datasets, we adopt CLIP as used by the authors of WinCLIP, following their zero-shot detection setup (not segmentation). The exact implementation details, including the prompts used, are provided in Appendix B.3.1 of our submission. Thus, in our results, the baseline CLIP performance reflects the standalone performance of WinCLIP. Our method shows the gains achieved when integrating CLIP-based evidence with a base AD model using EPHAD.
>
> Lastly, unlike MVFA, we do not fine-tune the CLIP model. To apply EPHAD in the medical domain, we can use any SOTA AD methods as the base detector with the finetuned CLIP or any other domain-specific evidence function.
>
> **[W4] The Introduction claims that existing methods rely on 'the contamination ratio.' However, these papers never mention such a ratio.** While it is true that many existing AD methods that operate under contamination do not explicitly assume knowledge of the exact contamination factor, they often rely, either directly or indirectly, on having a reasonable approximation of it for effective performance. For instance, in [2], the contamination factor $\alpha$ is treated as a hyperparameter that "characterizes the assumed fraction of anomalies" in the training data. The authors further provide a sensitivity analysis (Figure 3), which clearly demonstrates that performance degrades significantly, especially for LOE-Hard, when the assumed value deviates from the true contamination level. Similarly, in [3], the threshold $\gamma$ is introduced for refining the contaminated training set. In Section 3.3, the authors note that their method (SRR) remains robust as long as $\gamma$ is selected from a **reasonable range**, which they define as 1 to 2 times the true anomaly ratio. This again suggests that their method implicitly depends on a rough estimate of the true contamination level for optimal results. Therefore, while these works may not rely on the exact contamination factor, they do depend on having a reasonable approximation.
>
>
> [1] Patra et al., "Detecting Abnormal Operations in Concentrated Solar Power Plants from Irregular Sequences of Thermal Images." KDD, 2024.
> [2] Qiu et al., "Latent outlier exposure for anomaly detection with contaminated data." ICML, 2022.
> [3] Yoon et al., "Self-supervise, Refine, Repeat: Improving Unsupervised Anomaly Detection" TMLR, 2022.

---

> ### Author Response · Authors · 2025-08-05
>
> Dear Reviewer qm8P,
>
> Thank you once again for your time and insightful feedback, which has helped us improve our paper. We hope our rebuttal has addressed your concerns regarding the design of the evidence functions, including the validity of using CLIP, as well as our discussion of existing methods that rely on prior knowledge of the contamination ratio.
>
> If you have any further questions or comments, please let us know before the discussion period ends. We are happy to clarify any remaining points.
>
> Best regards,
> The Authors

---

> > ### Comment · Reviewer_qm8P · 2025-08-06
> >
> > Thank you for the response. The author avoided the data leakage issue of CLIP in rebuttal. Since CLIP can be directly used for zero shot anomaly detection in datasets such as MVTec without any fine-tuning to obtain SOTA results, using CLIP to filter abnormal data in the training set clearly results in unfair data leakage compared to other methods.
> > However, this is not what I am most concerned about. The main contribution of this paper is to provide a framework for handling data pollution. I am aware that the design of the Evidence Function is discussed in detail in the supplementary materials, but it is not mentioned at all in the main paper, which can easily lead readers to mistakenly believe that the Evidence Function is a general and simple function. In fact, the design of Evidence Functions is very complex, and the design will be completely different for different types of data, even requiring the use of very complex pre trained models.
> > Therefore, the author should include a brief discussion on the design of Evidence Functions in the main paper, such as how Evidence Functions are designed for different data types. Secondly, the author should openly acknowledge the data leakage risk of borrowing pre trained models and emphasize the role of such usage in improving the results. Appropriate explanations can prevent misunderstandings in subsequent research.

---

> > > ### Author Response · Authors · 2025-08-06
> > >
> > > Thank you for your comment. In the following, we provide a more detailed discussion of the concerns raised by the reviewer.
> > >
> > > **[W1] The author should openly acknowledge the data leakage risk of borrowing pre-trained models and emphasise the role of such usage in improving the results.** We acknowledge that leveraging large-scale pre-trained models such as CLIP introduces the potential for overlap between training data and evaluation benchmarks. In the era of foundational models, this is a common concern for any research that uses large pre-trained models such as CLIP or Large Language Models. We have revised the manuscript to more explicitly acknowledge this limitation and clarify the role of CLIP in our method. Specifically, we now clearly state that CLIP has been previously employed in zero-shot anomaly detection settings and is known to perform well on datasets such as MVTec and ViSA [1]. While this can offer a performance advantage, it is not intrinsic to EPHAD’s design. Our method is modular by construction, and CLIP is used solely as one of several possible evidence functions.
> > >
> > > To ensure transparency, we report the performance of stand-alone CLIP in Table 1, allowing readers to disentangle its contribution from the rest of the pipeline. Importantly, we also observe that in use-cases where the semantic alignment between CLIP’s training data and the target domain is weak, as in the Metal Parts Defect Dataset (MPDD) [2], the performance of CLIP degrades significantly. This in turn impacts the combined performance of EPHAD when the default weight ($\beta=0.5$) is used. However, by adaptively tuning $\beta$, we can reduce reliance on the evidence function and recover performance, which demonstrates the robustness and flexibility of our framework.
> > >
> > > Finally, we emphasise that EPHAD does not require a pre-trained vision-language model to function. We demonstrate its effectiveness on tabular and real-world datasets, where no such pre-trained models are used. This further underlines that EPHAD is a general framework and not inherently dependent on large-scale pretraining.
> > >
> > > We hope this clarifies our position and the measures taken to acknowledge and mitigate the risks associated with the use of a foundation model in our experiments.
> > >
> > > **[W2] I am aware that the design of the Evidence Function is discussed in detail in the supplementary materials, but it is not mentioned at all in the main paper, which can easily lead readers to mistakenly believe that the Evidence Function is a general and simple function.** Thank you for pointing this out. In fact, the design of the evidence function in EPHAD is not limited to simple heuristics and can range in complexity depending on the use case and data modality.
> > >
> > > While our initial draft deferred this discussion to the appendix due to space limitations, we acknowledge the reviewer’s concern that this may lead to a misunderstanding of the nature and role of the evidence function. We have revised the main paper to include important details about the evidence functions used in our paper for various data modalities. We will make the updated draft available as soon as we are allowed to upload it.
> > >
> > > [1] Jeong et al., "Winclip: Zero-/few-shot anomaly classification and segmentation." CVPR, 2023.
> > > [2] Jezek et al., "Deep learning-based defect detection of metal parts: evaluating current methods in complex conditions." ICUMT, 2021.

---

> > > > ### Comment · Reviewer_qm8P · 2025-08-07
> > > >
> > > > Thank you for the response. As the authors have addressed all my concerns, I will raise my score to accept.

---

> > > > > ### Author Response · Authors · 2025-08-07
> > > > >
> > > > > Thank you for updating the score. We are happy to note that our response addressed your concerns.

---

### Official Review · Reviewer_68bH · 2025-07-01

**Clarity:** 3
**Significance:** 2
**Originality:** 3
**Rating:** 5
**Confidence:** 4

**Summary:**

This work introduces a framework for correcting the posterior distribution of the inlier samples given a model that is trained with contaminated data. This framework, dubbed EPHAD, is based in a generalized Bayesian inference framework, using the a CLIP-based evidence function. EPHAD aims to perform test-time adaption by correcting the posterior via $p^c_{Y=+1} (x) = p^u_{Y=+1} (x) \exp (T(X)/\beta)$, where $p^u$ and $p^c$ are the uncorrected and corrected posterior probabilities. The work shows that using the EPHAD framework may help in improving the anomaly detection performance when the detectors are trained with contaminated data. Experimentation in tabular and image data is presented, with ablations for the introduced hyperparameters also shown. Additionally, the authors discuss the connection of their approach with inference-time alignment of generative models.

**Questions:**

- Why is it valid to use the entire test-set in one go as extra information to improve anomaly detection within the same test-set?

- How would you use this technique in a real-world scenario? Do you need a set of test images beforehand?

**Ethical Concerns:**

["NO or VERY MINOR ethics concerns only"]

**Final Justification:**

The authores answered most of my concerns and clarified about how they adapt TTA into anomaly detection within their framework. Additionally, the authors compromised to add further clarifications in the final version. I think the paper is of good value and has strong theoretical background, therefore I change my initial score from Reject to Accept. Good work!

**Limitations:**

Yes

**Quality:**

3

**Strengths And Weaknesses:**

I think the paper is really well written and has a strong mathematical background, with the appropiated works cited and discussed. The motivation behind it is quite sound, and I completely agree with the claims made by the authors in the introduction and related work. Additionally, there are several experiments including toy examples, tabular data and images, showing that the proposed approach improves the metrics.

However, I think I find a big flaw in the process that makes me go for a reject instead of an accept (which I would normally choose for a paper like this): your test-time adaption method is performed using the whole test set at the same time and then reporting the metrics using your adjustment framework. Specifically, you say: "In practice, the posterior is inferred from test samples, so the sample size n is constrained by the number of available test points." Basically, the problem I find is that in Equation (11), the $t$ is obtained from the test set and then used to adjust the posterior probability. From my point of view, that is training, because you are using the whole test-set to adjust the posterior to have a better performance on that test set. Therefore, the improved results for EPHAD are clearly obtained because you looked at the test set and then adjusted to perform better in that test set. I think it would be better if you had a validation set to perform and obtain the values for Eq (11), or if you implemented EPHAD in a batch manner. I am willing to change my rate if the problem I'm exposing is solved or justified.

Other minor weakneses:
- It'd be great to expand a bit more in Lines L183-L185
- L173: Should be Eq. (3), not (3)
- L195-196: How do you choose the threshold?
- Eq (13): I think you have an extra "exp"
- L291: I think the $\epsilon \\%$ should only be $\epsilon$
- I would expand a bit more in how did you use the evidence function for the image datasets and move the "Connection with inference-time alignment of generative models" section to the appendix.

---

> ### Author Rebuttal · Authors · 2025-07-31
>
> We sincerely appreciate the reviewer's thoughtful and constructive feedback. It is encouraging to hear that the reviewer finds our paper to be well written with clear motivation and a strong mathematical background. We address the reviewer's concerns in the points below.
>
> **[W1, Q1] Why is it valid to use the test-set as extra information to improve anomaly detection?** We would like to clarify a key aspect of our method, EPHAD, which may help address the concern regarding potential test set exposure.
>
> EPHAD is explicitly designed as a test-time adaptation (TTA) method in which anomaly scores produced by a detector trained on a contaminated dataset are updated during inference using an auxiliary evidence function. Importantly, **during adaptation, we do not have access to ground-truth labels for test samples. This ensures that our approach strictly avoids data leakage and remains consistent with standard practices in the TTA literature** [1,2].
>
> We would also like to highlight a critical difference between Source-Free Unsupervised Domain Adaptation (SF-UDA) and TTA. While both operate without test labels, SF-UDA typically involves a separate, offline adaptation phase before inference, often requiring iterative fine-tuning on the unlabelled validation samples from the test distribution [3]. In contrast, TTA methods, under which EPHAD falls, do not have a separate model adaptation step. Here, the model is adapted during inference. The benefits of TTA approaches are twofold: (i) they avoid iterative training, improving the computational efficiency, and (ii) they do not require additional validation samples, which is beneficial for data-starved settings such as AD.
>
> Moreover, EPHAD takes a step forward by being model-agnostic. It operates in a black-box setting, requiring only the output scores from the base detector. This flexibility broadens its applicability, particularly in settings where access to model internals is restricted.
>
> We hope this clarifies that the effectiveness of EPHAD does not arise from any exposure to the test set, but rather stems from our proposed adaptation strategy, aligned with established TTA methodologies.
>
> **[W2] It'd be great to expand a bit more in Lines L183-L185.** We have expanded lines 183-185 in our updated draft describing how EPHAD is a simple yet effective method for real-world applications. The revised text reads as follows:
>
> EPHAD is a simple yet effective inference-time adaptation framework aimed at mitigating the effects of training data contamination without requiring access to the pre-trained anomaly detection model’s weights or original training pipeline. This makes EPHAD uniquely suited for integration with existing systems, especially when retraining is not feasible due to resource, access, or time constraints. EPHAD can seamlessly incorporate evidence from foundation models such as CLIP and also domain-specific knowledge in specialised applications.
>
> **[W3] L173: Should be Eq. (3), not (3).** In the current draft, we have referred to equations using their corresponding number within the first bracket, e.g. (3). We have updated the format in our updated draft.
>
> **[W4] L195-196: How do you choose the threshold?** Threshold selection plays a crucial role in deciding the final label given an anomaly score. Thus, several existing studies are dedicated solely to this aspect [4-6]. However, the primary focus of our work lies in refining the anomaly scoring mechanism itself, not in optimizing the decision threshold. As such, determining a fixed threshold falls outside the scope of this study. To ensure a fair and threshold-independent evaluation aligned with our objectives, we report performance using AUROC.
>
> **[W5, W6] Eq (13): I think you have an extra "exp". L291: I think the $\epsilon\%$ should only be $\epsilon$.** We have fixed the typos in our updated draft.
>
> **[W7] I would expand a bit more in how did you use the evidence function.** Following the reviewer's suggestion and to help our readers, we added further details on how CLIP is used as an evidence function, which was previously in our appendix.
>
> **[Q2] How would you use this technique in a real-world scenario?** The approach presented in our current draft requires access to the full unlabelled test set to compute the outlier probability. Thus, it cannot be applied in online settings. However, we provide an alternative approach to applying EPHAD, which does not require converting the anomaly score to outlier probability for non-density-based methods. Consequently, EPHAD can be applied in an online setting, provided the evidence function can be computed sample-wise, such as when using CLIP or rule-based evidence functions. The alternative version of EPHAD is summarised below:
>
> Given an AD model trained on the contaminated dataset, we denote the anomaly score for a data point $x$ as $ s^-_u(x) $. Then, we can compute the inlier score as $ s^+_u(x) = - s^-_u(x) $. Considering $ s^+_u(x) $ as prior and an auxiliary evidence function $T(x)$, EPHAD computes the revised score $ s^+_c(x) $ using exponential tilting. It is a technique used to adjust a PDF by ``tilting'' it toward a specific outcome. Recall that the inlier score is an order-preserving transformation of the inlier PDF, i.e., $ s^+(x) = \phi(f^+_X(x)) $ where $\phi$ is a transformation function, e.g. log-likelihood. Thus, given a score-based AD model that learns $s^-_u(x)$, tilting increases the relative scores of the normal samples over the anomalous samples, steering the model toward an outcome supported by the evidence function. We first exponentiate the inlier score $ s^+_u(x) = - s^-_u(x) $ to ensure non-negativity. Then, we rescale $T(x)$ with a temperature parameter $\beta$ and exponentiate it. Finally, the revised inlier score is computed as:
>
> $$s^+_c(x) = \exp(s^+_u(x))\exp(T(x)/\beta).$$
>
> [1] Wang et al., "Tent: Fully test-time adaptation by entropy minimization." ICLR 2021.
> [2] Xiao et al., "Beyond model adaptation at test time: A survey." ArXiv 2024.
> [3] Fang et al., "Source-free unsupervised domain adaptation: A survey." Neural Networks 2024.
> [4] Perini et al., "Estimating the contamination factor’s distribution in unsupervised anomaly detection." ICML, 2023.
> [5] Perini et al., "Transferring the contamination factor between anomaly detection domains by shape similarity.", AAAI, 2022.
> [6] Hendrickx et al., "Machine learning with a reject option: a survey." Machine Learning, 2024.

---

> ### Author Response · Authors · 2025-08-05
>
> Dear Reviewer 68bH,
>
> Thank you once again for your time and insightful feedback, which has helped us improve our paper. We hope our rebuttal has addressed your concerns regarding the validity of our approach and the application of EPHAD in real-world scenarios.
>
> If you have any further questions or comments, please let us know before the discussion period ends. We are happy to clarify any remaining points.
>
> Best regards,
> The Authors

---

> ### Comment · Reviewer_68bH · 2025-08-05
>
> Dear authors,
>
> Thanks for the detailed rebuttal. You response does answer my main concerns. I appreciate your more detailed explanation about the settings of your method and I believe is a moderate to good addition to the anomaly detection literature. Therefore, I'll update my score from Reject to Accept, in light to not fall into any Borderline option. I think once concern I still have is the one raised by reviewer qm8P about CLIP data leakage, so I'd recommend to go deeper in the discussion in the final paper. In any case, it is a good rebuttal, well done!

---

> > ### Author Response · Authors · 2025-08-06
> >
> > Thank you for updating the score. We are happy to note that our response addressed your main concerns. In the following, we provide a more detailed discussion of the concern regarding the CLIP usage.
> >
> > **[W1] The data leakage risk when using CLIP and the role of such usage in improving the results.**
> >
> > We acknowledge that leveraging large-scale pre-trained models such as CLIP introduces the potential for overlap between training data and evaluation benchmarks. In the era of foundational models, this is a common concern for any research that uses large pre-trained models such as CLIP or Large Language Models. We have revised the manuscript to more explicitly acknowledge this limitation and clarify the role of CLIP in our method. Specifically, we now clearly state that CLIP has been previously employed in zero-shot anomaly detection settings and is known to perform well on datasets such as MVTec and ViSA [1]. While this can offer a performance advantage, it is not intrinsic to EPHAD’s design. Our method is modular by construction, and CLIP is used solely as one of several possible evidence functions.
> >
> > To ensure transparency, we report the performance of stand-alone CLIP in Table 1, allowing readers to disentangle its contribution from the rest of the pipeline. Importantly, we also observe that in use-cases where the semantic alignment between CLIP’s training data and the target domain is weak, as in the Metal Parts Defect Dataset (MPDD) [2], the performance of CLIP degrades significantly. This in turn impacts the combined performance of EPHAD when the default weight ($\beta=0.5$) is used. However, by adaptively tuning $\beta$, we can reduce reliance on the evidence function and recover performance, which demonstrates the robustness and flexibility of our framework.
> >
> > Finally, we emphasise that EPHAD does not require a pre-trained vision-language model to function. We demonstrate its effectiveness on tabular and real-world datasets, where no such pre-trained models are used. This further underlines that EPHAD is a general framework and not inherently dependent on large-scale pretraining.
> >
> > We hope this clarifies our position and the measures taken to acknowledge and mitigate the risks associated with the use of a foundation model in our experiments.
> >
> > [1] Jeong et al., "Winclip: Zero-/few-shot anomaly classification and segmentation." CVPR, 2023.
> > [2] Jezek et al., "Deep learning-based defect detection of metal parts: evaluating current methods in complex conditions." ICUMT, 2021.

---

### Official Review · Reviewer_sGXx · 2025-07-02

**Clarity:** 3
**Significance:** 3
**Originality:** 3
**Rating:** 4
**Confidence:** 3

**Summary:**

This paper addresses the problem of deploying unsupervised anomaly detection (AD) models that have been trained on datasets contaminated with anomalies without the possibility of retraining, modifying the model, or accessing the original training data. To tackle this, the authors propose EPHAD, a general post-hoc framework that adjusts model outputs at inference time using auxiliary evidence. The approach is formulated as a Bayesian correction, treating the original AD model as a prior and integrating external evidence (e.g., outputs from foundation models like CLIP) as likelihood signals. EPHAD supports both density-based and score-based AD models by estimating inlier probabilities in a statistically principled way using test-time rankings and a Beta posterior.

**Questions:**

1. Can EPHAD be adapted for online settings? The score-based update requires estimating the empirical score distribution across the test batch. Is it feasible to apply EPHAD when samples arrive sequentially, and if so, how could the rank-based prior be computed incrementally?

2. Why are standard deviations not reported in Table 2?

3. What is the computational cost of EPHAD at inference time, especially when using large evidence models like CLIP?

4. How is the decision threshold λ selected in the absence of labeled validation data?

5. How does EPHAD handle cases where the evidence function T(x) is noisy? How robust is EPHAD if the evidence is misleading?

**Ethical Concerns:**

["NO or VERY MINOR ethics concerns only"]

**Final Justification:**

I initially viewed this paper positively for its practical and well-motivated framework for post-hoc anomaly detection correction, but raised questions about its applicability in online settings, threshold selection, the inference-time cost of large evidence models such as CLIP, and robustness when the auxiliary evidence function T(x) is noisy or misaligned. The authors’ rebuttal addressed these questions, providing explanations that clarified the method’s design and deployment considerations.
While these responses resolved my specific questions, they did not materially change my overall assessment. The strengths of the paper remain in its broad applicability, principled formulation, and strong empirical results, while the key limitations, such as reliance on evidence quality and lack of calibration evaluation, remain. I therefore keep my original evaluation unchanged.

**Limitations:**

- The score-based variant of EPHAD requires access to a batch of test-time scores for rank-based estimation, which makes it unsuitable for online AD scenarios.

- The framework depends on auxiliary evidence functions, whose quality significantly affects performance. Though some mitigation strategies are discussed, the method assumes that the evidence is broadly aligned with the true anomaly structure.

- Although the method is designed for test-time use, some evidence functions such as CLIP may introduce non-trivial computational overhead.

**Paper Formatting Concerns:**

No concerns.

**Quality:**

3

**Strengths And Weaknesses:**

Strengths:

- The paper addresses an important but underexplored real-world challenge: adapting AD models trained on contaminated data when retraining is not feasible. This reflects realistic deployment scenarios, such as in industry, where model internals and training data are often unavailable.

- The paper draws a novel connection between post-hoc anomaly score correction and KL-regularized inference, as used in reward shaping for generative models and policy improvement in reinforcement learning (RL).

- EPHAD’s use of auxiliary evidence functions offers flexibility across domains and modalities which makes it agnostic to the source of evidence and helps to incorporate external knowledge without modifying the base AD model.

- The experiments convincingly show that EPHAD improves performance under contamination across diverse AD models, data modalities, and domains.

Weaknesses:

- **Thresholding Left Unspecified:** The paper uses thresholds λp and λs to make binary decisions but does not describe how these are chosen.

- **No reporting of standard deviations in result table:** In Table 2 (tabular datasets), only mean AUROC scores are reported, with no variance across seeds.

- **Lack of resource efficiency reporting or runtime metrics:** Despite promoting EPHAD as a practical test-time intervention, the paper provides no analysis of runtime overhead, memory usage, or latency—particularly when computationally expensive evidence functions like CLIP are used.

- **Missing discussion of broader societal impacts:** The authors mark the broader impact section as “NA” without justification, despite the method being applicable to high-stakes domains such as industrial automation, medical diagnostics, or financial fraud detection.

---

> ### Author Rebuttal · Authors · 2025-07-31
>
> We sincerely appreciate the reviewer’s insightful and valuable feedback. We are happy to note that the reviewer finds the problem addressed in our work to be an important but under-explored real-world challenge. We address the reviewer's concerns in the points below.
>
> **[W1, Q4] Thresholding Left Unspecified.** Selecting a threshold is an essential consideration in deciding the final label given an anomaly score. Thus, several existing studies are dedicated solely to this aspect [1-3]. However, the primary focus of our work lies in refining the anomaly scoring mechanism itself, not in optimising the decision threshold. As such, determining a fixed threshold falls outside the scope of this study. To ensure a fair and threshold-independent evaluation aligned with our objectives, we report performance using AUROC.
>
> **[W2, Q2] No reporting of standard deviations in result table.** We have now repeated all the experiments with three seeds in our updated draft.
>
> **[W3, Q3, L3] Lack of resource efficiency reporting or runtime metrics.** We would like to emphasise that EPHAD operates without modifying the base AD model and does not introduce any additional fine-tuning steps. As a result, the memory and computational requirements remain unchanged relative to the original model. The only additional cost stems from the auxiliary evidence function, whose overhead depends on the specific implementation chosen.
>
> For example, in our real-world use case involving solar power plants, a simple rule-based heuristic serves as the evidence function, resulting in negligible memory and compute overhead. Similarly, in the tabular AD setting where we employ Local Outlier Factor (LOF), the added cost is minimal. In scenarios involving foundation models such as CLIP, while there is an increase in memory usage due to model size, the latency remains low, as adaptation involves only a single forward pass followed by a lightweight similarity computation with predefined templates.
>
> To better illustrate these trade-offs, we will include a dedicated section in the updated draft of our paper that reports runtime and memory overheads across the different use cases under consideration to provide clarity about the practical applicability of EPHAD.
>
> **[W4] Missing discussion of broader societal impacts.** Thank you for your suggestion. We will incorporate a section discussing the broader impact of our work in the updated draft. EPHAD provides an effective way to adapt pre-trained AD models during inference, removing the need for specialised architectures to handle data contamination. This is crucial for high-stakes applications such as industrial automation and medical diagnostics.
>
> **[Q1, L1] Can EPHAD be adapted for online settings?** For the approach discussed in our current draft, we need to have the unlabelled test set together to compute the outlier probability. Thus, it cannot be applied in online settings. However, we provide an alternative approach to applying EPHAD, which does not require converting the anomaly score to outlier probability for non-density-based methods. Consequently, EPHAD can be applied in an online setting, provided the evidence function can be computed sample-wise, such as when using CLIP or rule-based evidence functions. The alternative version of EPHAD is summarised below:
>
> Given an AD model trained on the contaminated dataset, we denote the anomaly score for a data point $x$ as $ s^-_u(x) $. Then, we can compute the inlier score as $ s^+_u(x) = - s^-_u(x) $. Considering $ s^+_u(x) $ as prior and an auxiliary evidence function $T(x)$, EPHAD computes the revised score $ s^+_c(x) $ using exponential tilting. It is a technique used to adjust a PDF by ``tilting'' it toward a specific outcome. Recall that the inlier score is an order-preserving transformation of the inlier PDF, i.e., $ s^+(x) = \phi(f^+_X(x)) $ where $\phi$ is a transformation function, e.g. log-likelihood. Thus, given a score-based AD model that learns $s^-_u(x)$, tilting increases the relative scores of the normal samples over the anomalous samples, steering the model toward an outcome supported by the evidence function. We first exponentiate the inlier score $ s^+_u(x) = - s^-_u(x) $ to ensure non-negativity. Then, we rescale $T(x)$ with a temperature parameter $\beta$ and exponentiate it. Finally, the revised inlier score is computed as:
>
> $$s^+_c(x) = \exp(s^+_u(x))\exp(T(x)/\beta).$$
>
> **[Q5, L2] How does EPHAD handle cases where the evidence function $T(x)$ is noisy? How robust is EPHAD if the evidence is misleading?** In our experiments on the image-based AD datasets presented in Table 1, we employed CLIP as the evidence function. In certain datasets, such as SVHN, the evidence function can be noisy. For instance, when used in isolation (i.e., without EPHAD), CLIP achieves an AUROC of only 58.46\% on SVHN, indicating limited reliability in this context. In such cases, applying EPHAD with the default setting of $\beta=0.5$ does not lead to improvements over the base detector. However, we emphasise that $\beta$ is a tunable parameter.  As shown in Figure 4 in our current draft, EPHAD offers a smooth transition between the two models, illustrating not only reliability but also adaptability, offering a principled way to balance the contributions of the base detector and the evidence function. To further support this adaptability, we provide an unsupervised strategy for selecting an appropriate $\beta$ in Appendix C.6.
>
> [1] Perini et al., "Estimating the contamination factor’s distribution in unsupervised anomaly detection." ICML, 2023.
> [2] Perini et al., "Transferring the contamination factor between anomaly detection domains by shape similarity.", AAAI, 2022.
> [3] Hendrickx et al., "Machine learning with a reject option: a survey." Machine Learning, 2024.

---

> > ### Comment · Reviewer_sGXx · 2025-08-04
> > **Thank you !**
> >
> > I appreciate the authors’  response to my earlier concerns. The additional explanations address the main issues I had raised. At this stage, I do not have further questions. In light of the authors’ response, I believe my initial assessment remains appropriate, and I will keep my score unchanged.

---

> > > ### Author Response · Authors · 2025-08-06
> > >
> > > Thank you for your time. We are glad to note that our response addressed your main concerns.

---

### Decision · Program_Chairs · 2025-09-17

**Decision:**

Accept (spotlight)

**Comment:**

This paper addresses the important problem of anomaly detection under contaminated training data by introducing a novel Bayesian test-time adaptation framework, demonstrating strong versatility and effectiveness across diverse models and datasets. Its strengths lie in the practical motivation, sound methodology, and comprehensive experiments confirming consistent improvements. However, reviewers raised concerns about threshold selection, computational efficiency, evidence function design, and the use of the test set, questioning some aspects of the empirical validity. The rebuttal successfully addressed most concerns, and all reviewers expressed positive evaluations. I recommend acceptance, with the expectation that final revisions incorporate reviewer feedback.